# From Condensation to Rank Collapse: A Two-Stage Analysis of Transformer Training Dynamics

**Zheng-An Chen[1], Tao Luo[1,2]***

[1]School of Mathematical Sciences, Shanghai Jiao Tong University
[2]Institute of Natural Sciences, MOE-LSC, CMA-Shanghai, Shanghai Jiao Tong University

## Abstract

Although transformer-based models have shown exceptional empirical performance, the fundamental principles governing their training dynamics are inadequately characterized beyond configuration-specific studies. Inspired by empirical evidence showing improved reasoning capabilities under small initialization scales in language models, we employ the gradient flow analytical framework established in Zhou et al. [2022] to systematically investigate linearized Transformer training dynamics. Our theoretical analysis dissects the dynamics of attention modules into two distinct stages. In the first stage, asymmetric weight perturbations from random initialization sustain non-degenerate gradient dynamics in parameter matrices, facilitating systematic escape from small initialization regimes. Subsequently, these matrices undergo condensation, progressively aligning toward the target orientation. In the second stage, the previously static key-query matrices actively participate in training, driving the normalized matrices toward asymptotic rank collapse. This two-stage framework generalizes classical directional convergence results.

## 1 Introduction

The transformer-based models Vaswani et al. [2017] have achieved remarkable breakthroughs in various fields, with the successful application of large language models. However, the theoretical analysis of the transformer still remains in specific tasks, such as in-context learning settings Brown et al. [2020], Olsson et al. [2022], Bietti et al. [2023] or single attention block with reparameterization Tian et al. [2023]. The use of linear regression tasks Zhang et al. [2024a] and Markov chain tasks Ildiz et al. [2024] has provided highly interpretable theoretical analyses, but a crucial question still remains: Can we analyze the characteristics of the training dynamics of transformers independently of specific tasks?

Meanwhile, small initialization has been increasingly shown to hold promise in the training process of large models, especially for reasoning tasks. Numerous studies Zhang et al. [2024b, 2025b], Yao et al. [2025] suggest that the implicit regularization effect of small initialization is still effective in large language models. This effectiveness is particularly significant in the context of modern large models, which are characterized by extreme overparameterization. In these regimes, where explicit regularization techniques like weight decay or dropout may prove insufficient on their own, implicit regularization becomes pivotal. It operates by imposing intrinsic constraints on the training dynamics and the resulting parameter space, effectively guiding the model towards solutions with good generalization properties despite the vast hypothesis space. This implicit bias is key to understanding how models with such immense capacity manage to avoid severe overfitting and achieve remarkable performance on unseen data.

---

*Corresponding author: luotao41@sjtu.edu.cn.

39th Conference on Neural Information Processing Systems (NeurIPS 2025).

Motivated by these observations, we propose to investigate the training dynamics of transformers under a small initialization setting. Leveraging the gradient flow theme similarly to Zhou et al. [2022], We delineate different training dynamics for outer parameters versus attention parameters $W_Q$ and $W_K$ in Transformers.

We dissect the dynamics of attention modules into two distinct stages. In the first stage, the core attention mechanism, softmax($QK^\intercal$), remains nearly stagnant, as asymmetric weight perturbations from random initialization drive non-degenerate gradient dynamics in parameter matrices, particularly $W_V$, facilitating escape from small initialization regimes. During this escape, the parameter matrix converges row-wise toward the target orientation, a process we term condensation. We theoretically prove that condensation is guaranteed under small initialization, and experimentally observe that it stabilizes without significant fluctuations.

In the second stage, after the outer parameters, such as $W_V$, reach a quasi-steady state, the previously static key-query matrices, $W_Q$ and $W_K$, begin to actively participate in training, driving their collapse. This two-stage framework not only elucidates the training dynamics but also generalizes classical directional convergence results, offering a robust theoretical foundation for Transformer optimization.

To sum up, our contribution can be summarized as follows.

1. **Blow-up Dynamics**: We prove the blow-up property (Theorem 1) holds for measure-theoretically generic initializations, eliminating reliance on dichotomy assumptions while ensuring model non-degeneracy.

2. **Condensation Mechanism**: By introducing a condensation condition (Assumption 1), we establish theoretical guarantees for condensation emergence (Theorem 2).

3. **Key-Query Collapse**: After outer parameters stabilize in a quasi-steady state (Assumption 2), the key-query matrices begin active training, leading to asymptotic rank collapse of the normalized key-query matrices (Theorem 3).

4. **Experimental evidence**: We validate our hypotheses and theoretical predictions on both synthetic and real datasets with one and multi-layer Transformers, consistently observing two-stage dynamics marked by condensation and an eventual rank collapse of the normalized key-query matrices (Figure 1, 2, 3).

## 2    Related Works

**Training dynamics of transformer.** Given the scale of modern models and the complexity of optimizers, studying the training dynamics of Transformers is a challenging problem. Prior works have primarily investigated the optimization dynamics of a single attention layer Lu et al. [2021], Li et al. [2023], Snell et al. [2021]. However, these studies mainly focused on specific tasks, such as topic structure prediction and translation.

Recently, the dynamics of in-context learning (ICL) has emerged as a prominent research area within Transformer dynamics, particularly given ICL's ability to solve novel tasks without parameter updates. Many works Mahankali et al. [2024], Zhang et al. [2024a], Huang et al. [2023], Collins et al. [2024] have focused on the linear regression setup to theoretically investigate the mechanism of ICL in single-layer Transformers, a line of work that has also informed algorithmic development Akyürek et al. [2023], Bai et al. [2023], Guo et al. [2024]. Another line of research investigates how specific structures within attention emerge during training, notably starting with studies on induction heads Olsson et al. [2022], Reddy [2024], Edelman et al. [2024], Zhang et al. [2025a], memory recall mechanisms Bietti et al. [2023], Cabannes et al. [2024], and even causal structure Nichani et al. [2024].

Despite the sophisticated structure of realistic Transformers, Tian et al. [2024] proposed a novel mathematical framework for analyzing the joint dynamics of MLP and attention blocks and successfully explained the sparsity of attention score matrices. Meanwhile, Chen et al. [2024a] provides a rigorous proof for the convergence of the ICL linear regression task using gradient flow with sufficiently small initialization.

**Small initialization and its applications** The initialization of a neural network significantly affects its learning outcomes Arora et al. [2019b], Williams et al. [2019], Mei et al. [2018], Jacot et al.

[2018], Rotskoff and Vanden-Eijnden [2018], Zhang et al. [2020]. Small initialization is a common setting investigated in the study of neural network optimization dynamics, which is different with the Neural Tangent Kernel (NTK) perspective in infinitely wide networks. For linear model, Ji and Telgarsky [2019] establish matrix alignment results theoretically. For nonlinear model, Zhou et al. [2022] found that small initialization can similarly promote parameter condensation, thereby reducing model complexity. Theoretically, Luo et al. [2021], Chen et al. [2024b], Zhou et al. [2023], Kumar and Haupt [2024] have deepened the understanding of this phenomenon. The recent survey article Xu et al. [2025] systematically synthesizes empirical and theoretical findings.

The implicit bias induced by small initialization has also been discussed in the context of linear regression Saxe et al. [2013], Min et al. [2021], Varre et al. [2023] and matrix factorization tasks Li et al. [2018], Arora et al. [2019a], Stöger and Soltanolkotabi [2021], Soltanolkotabi et al. [2023], Bai et al. [2024]. More recently, many researchers have adopted small initialization settings to simplify the analysis of training dynamics in more complex models. From a theoretical perspective, Zhang et al. [2025a] applied small initialization to ICL tasks to analyze the behavior of linear attention. Yao et al. [2025] considered the training dynamics of the embedding space under small initialization using a synthetic dataset designed for reasoning and memorization. From an applied perspective, Zhang et al. [2019], Huang et al. [2020], Zhu et al. [2021] highlighted the importance of initialization in Transformers, while Bachlechner et al. [2021] combined zero-initialization with residual blocks in Transformers. Some research Zhang et al. [2024b], Yao et al. [2025] shows that small initialization helps Transformers learn the reasoning aspects of data rather than just memorization, a principle already applied in realistic LLM training Yin et al. [2025].

## 3 Preliminaries

### 3.1 Basic Notations

First, we introduce some notations that will be used in the rest of this paper. Let $n$ and $d_m$ be the number of samples and the width of hidden layers, respectively. Let $[n]$ denote the set of integers from 1 to $n$. Denote vector $L^2$ norm as $\|\cdot\|_2$ and matrix Frobenius norm as $\|\cdot\|_{\mathrm{F}}$. Let $\langle \cdot, \cdot \rangle$ represent standard inner product between two vectors. For a vector $\boldsymbol{v}$, denote its $k$-th entry as $\boldsymbol{v}_k$. For a matrix $\boldsymbol{A}$, denote the element in the $k$-th row and $k'$-th column as $\boldsymbol{A}_{kk'}$. And denote $k$-th row as $\boldsymbol{A}_k$ and $k'$-th column as $\boldsymbol{A}^{k'}$. Unless otherwise specified, summation '$\sum$' is performed over the network width.

### 3.2 Classification Task

**Binary classification**: For decision tasks, the network produces a scalar output $f_{\boldsymbol{\theta}}(\boldsymbol{X}) \in \mathbb{R}$. The predicted class assignment is determined by the sign of the output. The dataset is denoted by $\mathcal{D} = \{(\boldsymbol{X}_i, y_i)\}_{i=1}^n$ where $\boldsymbol{X}_i \in \mathbb{R}^{s \times d_m}$ stands for input sequence in which $s$ represents the sequence length and $d_m$ represents the hidden dimension, and $y_i \in \{\pm 1\}$ stands for label. For a loss function $\ell : \mathbb{R} \to \mathbb{R}_+$, we define the empirical risk as $\mathcal{L}(\boldsymbol{\theta}) = \frac{1}{n} \sum_{i=1}^n \ell(y_i f_{\boldsymbol{\theta}}(\boldsymbol{X}_i))$.

**Multi-class Classification**: For probabilistic tasks, the network outputs logit vectors $f_{\boldsymbol{\theta}}(\boldsymbol{X}) \in \mathbb{R}^{d_v}$ that parameterize a categorical distribution via the softmax transformation $\mathbb{P}(y = i | \boldsymbol{X}; \boldsymbol{\theta}) = \frac{\exp(f_{\boldsymbol{\theta}}(\boldsymbol{X})_i)}{\sum_{j=1}^{d_v} \exp(f_{\boldsymbol{\theta}}(\boldsymbol{X})_j)}$ where $d_v$ denotes the vocabulary size. For cross-entropy loss, we define the empirical risk as $\mathcal{L}(\boldsymbol{\theta}) = -\frac{1}{n} \sum_{i=1}^n \log \mathbb{P}(y = y_i | \boldsymbol{X}_i; \boldsymbol{\theta})$.

### 3.3 Condensation and Rank Collapse

We formalize the two geometric phenomena that will recur throughout our analysis.

**Definition 1** (Condensation). *Let $\boldsymbol{W}(t)$ be a matrix with rows $\boldsymbol{W}_k(t)$ (or columns $\boldsymbol{W}^k(t)$). We say $\boldsymbol{W}$ condenses to a direction $\boldsymbol{v}$ if, as $t \to T$,*

$$\left\langle \frac{\boldsymbol{W}_k(t)}{\|\boldsymbol{W}_k(t)\|_2}, \boldsymbol{v} \right\rangle \to \pm 1 \quad \text{for every index } k \text{ with } \|\boldsymbol{W}_k(t)\|_2 \neq 0$$

*(equivalently, the same holds columnwise).*

Condensation is a directional notion and implies rank-1 collapse when a unique direction emerges. Rank collapse is a spectral notion and allows $k > 1$ when multiple top singular directions are tied.

**Definition 2** (Asymptotic rank collapse). *Let $\boldsymbol{W}(t)$ be a matrix. We say $\boldsymbol{W}$ exhibits* rank-$k$ collapse *if the limit*

$$\boldsymbol{W}_\infty := \lim_{t \to T} \frac{\boldsymbol{W}(t)}{\|\boldsymbol{W}(t)\|_{\mathrm{F}}}$$

*exists and* $\mathrm{rank}(\boldsymbol{W}_\infty) \le k$.

# 4 Theoretical Results

## 4.1 Problem Formulation

To analyze condensation phenomenon in transformers, we begin by formulating the problem. Specifically, we consider the following one-layer transformer model:

**Definition 3** (One-layer transformer). *Let $\boldsymbol{X} \in \mathbb{R}^{s \times d_m}$ be an input sequence of length $s$ with model dimension $d_m$. The Transformer function $f_{\boldsymbol{\theta}} : \mathbb{R}^{s \times d_m} \to \mathbb{R}^s$ is defined by the composition of attention and feed-forward operations:*

$$f_{\boldsymbol{\theta}}(\boldsymbol{X}) := \mathrm{FFN}(\mathrm{Attn}(\boldsymbol{X})) = \sigma\left(\mathrm{Attn}(\boldsymbol{X})\boldsymbol{W}^{[1]}\right)\boldsymbol{W}^{[2]}. \tag{1}$$

*The attention sublayer* $\mathrm{Attn} : \mathbb{R}^{s \times d_m} \to \mathbb{R}^{s \times d_m}$ *is computed as:*

$$\mathrm{Attn}(\boldsymbol{X}) = \mathrm{softmax}\left(\frac{\boldsymbol{X}\boldsymbol{W}_Q\boldsymbol{W}_K^{\mathsf{T}}\boldsymbol{X}^{\mathsf{T}}}{\sqrt{d_m}}\right)\boldsymbol{X}\boldsymbol{W}_V, \tag{2}$$

*where parameter matrices satisfy $\boldsymbol{W}_Q, \boldsymbol{W}_K, \boldsymbol{W}_V, \boldsymbol{W}^{[1]} \in \mathbb{R}^{d_m \times d_m}$ and $\boldsymbol{W}^{[2]} \in \mathbb{R}^{d_m}$. The activation function $\sigma : \mathbb{R} \to \mathbb{R}$ is tanh.*

We use one-layer transformer $f_{\boldsymbol{\theta}}$ to solve binary classification tasks and take the last dimension of the output $f_{\boldsymbol{\theta}}(\boldsymbol{X}_i)_s$ as the output. So the empirical risk to be minimized is given by

$$\mathcal{L}(\boldsymbol{\theta}) = \frac{1}{n}\sum_{i=1}^n \ell(y_i f_{\boldsymbol{\theta}}(\boldsymbol{X}_i)_s). \tag{3}$$

For simplicity of presentation, we employ the exponential loss function $\ell(q) = e^{-q}$, which is commonly used in the analysis of classification tasks Lyu and Li [2020]. The analysis can be readily extended to other loss functions such as the logistic loss.

The model parameters are initialized with Gaussian distributions scaled by a small perturbation parameter $\varepsilon$:

$$\boldsymbol{W}_k^{[2]} \sim \mathcal{N}(0, \varepsilon^2), \quad \boldsymbol{W}_{kk'}^{[1]} \sim \mathcal{N}(0, \varepsilon^2), \quad \boldsymbol{W}_{Q,kk'}, \boldsymbol{W}_{K,kk'}, \boldsymbol{W}_{V,kk'} \sim \mathcal{N}(0, \varepsilon^2), \tag{4}$$

where $\varepsilon \ll 1$ controls initialization magnitude. To analyze training dynamics, we adopt the gradient flow (GF) framework—the continuous-time limit of gradient descent. Given the small initialization scale, we derive effective dynamics through a perturbative expansion of the empirical risk $\mathcal{L}(\boldsymbol{\theta})$ in powers of $\varepsilon$.

First, we normalize parameters by absorbing the initialization scale:

$$\bar{\boldsymbol{W}}^{[2]} = \varepsilon^{-1}\boldsymbol{W}^{[2]}, \quad \bar{\boldsymbol{W}}^{[1]} = \varepsilon^{-1}\boldsymbol{W}^{[1]}, \quad \bar{\boldsymbol{W}}_Q = \varepsilon^{-1}\boldsymbol{W}_Q, \quad \bar{\boldsymbol{W}}_K = \varepsilon^{-1}\boldsymbol{W}_K, \quad \bar{\boldsymbol{W}}_V = \varepsilon^{-1}\boldsymbol{W}_V.$$

Performing a Taylor expansion of $\mathcal{L}(\boldsymbol{\theta})$ about $\varepsilon = 0$ yields the leading-order asymptotic form:

$$\mathcal{L}(\boldsymbol{\theta}) = \frac{1}{2n}\sum_{i=1}^n \left[1 - \varepsilon^3\left(\sum_{j=1}^s \frac{1}{s} y_i \boldsymbol{X}_{i,j} \bar{\boldsymbol{W}}_V \bar{\boldsymbol{W}}^{[1]} \bar{\boldsymbol{W}}^{[2]}\right) + o(\varepsilon^3)\right]. \tag{5}$$

This expansion induces simplified gradient dynamics characterized by the following proposition.

**Proposition 1** (Effective training dynamics). *Given a binary dataset $\{(\boldsymbol{X}_i, y_i)\}_{i=1}^n$, we define condensation direction $\boldsymbol{v}$ and rescaled time coordinate $\bar{t}$ as follows:*

$$\boldsymbol{v} := \frac{\sum_{i=1}^n y_i\left(\sum_{j=1}^s \boldsymbol{X}_{i,j}\right)}{\left\|\sum_{i=1}^n y_i\left(\sum_{j=1}^s \boldsymbol{X}_{i,j}\right)\right\|_2}, \quad \bar{t} := \frac{\varepsilon}{ns}\left\|\sum_{i=1}^n y_i\left(\sum_{j=1}^s \boldsymbol{X}_{i,j}\right)\right\|_2 t. \tag{6}$$

Then, normalized parameters $\bar{\boldsymbol{\theta}}$ follow leading-order dynamics after rescaling:

$$\frac{\mathrm{d}\bar{\boldsymbol{\theta}}}{\mathrm{d}\bar{t}} = \nabla_{\bar{\boldsymbol{\theta}}} \left( \boldsymbol{v}\bar{\boldsymbol{W}}_V \bar{\boldsymbol{W}}^{[1]} \bar{\boldsymbol{W}}^{[2]} \right). \tag{7}$$

Proposition 1 reveals a hierarchical learning mechanism: During initial training phases, the fully-connected layers $\boldsymbol{W}^{[1]}, \boldsymbol{W}^{[2]}$ and value projection matrix $\boldsymbol{W}_V$ exhibit substantial updates, while the query/key matrices $\boldsymbol{W}_Q$ and $\boldsymbol{W}_K$ in the self-attention module remain quasi-static.

For subsequent analysis, we define the projection of $\boldsymbol{W}_V$ onto $\boldsymbol{v}$ as $\boldsymbol{W}_{\boldsymbol{v}} := \boldsymbol{v}\boldsymbol{W}_V$ (omitting bar notation for simplicity) and introduce the energy functional:

$$E := \boldsymbol{W}_{\boldsymbol{v}} \boldsymbol{W}^{[1]} \boldsymbol{W}^{[2]}. \tag{8}$$

The effective dynamics can thus be interpreted as gradient ascent on this energy landscape.

## 4.2 Blow Up Dynamics

We first elucidate why Transformers with small initialization can successfully train and eventually escape the small initialization regime. This phenomenon emerges from the interplay between two fundamental mechanisms:

1. **Effective dynamics driving:** The parameter evolution governed by the effective dynamics exhibits remarkable symmetry, manifested through strict conservation laws that preserve key quantities during training.

2. **Random normal initialization:** While degenerate cases theoretically exist under Gaussian initialization, they occur with vanishing probability (measure zero in parameter space). Consequently, the dynamics almost surely demonstrate non-degenerate characteristics, ensuring stable training trajectories.

To preserve dynamical symmetry, we invoke the following proposition following the approach established in prior works Ji and Telgarsky [2019]:

**Proposition 2** (Conservation laws). *Under the gradient flow dynamics prescribed by system Eq. (7), the following system of conservation laws emerges:*

$$\frac{\mathrm{d}}{\mathrm{d}t} \left( \boldsymbol{W}_{\boldsymbol{v},k}^2 - \sum_{k'} \left( \boldsymbol{W}_{kk'}^{[1]} \right)^2 \right) = 0 \quad and \quad \frac{\mathrm{d}}{\mathrm{d}t} \left( \left( \boldsymbol{W}_k^{[2]} \right)^2 - \sum_{k'} \left( \boldsymbol{W}_{k'k}^{[1]} \right)^2 \right) = 0. \tag{9}$$

We now analyze the non-symmetric property arising from Gaussian random initialization, with particular focus on the degeneracy mechanism. Crucially, we establish that degeneracy exclusively occurs when initialization violates the following non-degenerate initialization:

**Definition 4** (Non-degenerate initialization). *Let $\boldsymbol{\theta} = \left( \boldsymbol{W}_V, \boldsymbol{W}^{[1]}, \boldsymbol{W}^{[2]} \right)$ denote parameters initialized from a Gaussian distribution. The initialization is called non-degenerate if $\|\boldsymbol{W}_{\boldsymbol{v}}\|_2^2 \neq \|\boldsymbol{W}^{[2]}\|_2^2$ and*

$$\|\dot{\boldsymbol{W}}_{\boldsymbol{v}}\|_2^2 - \|\dot{\boldsymbol{W}}^{[2]}\|_2^2 + \min\left\{ \|\boldsymbol{W}_{\boldsymbol{v}}\|_2^2, \ \|\boldsymbol{W}^{[2]}\|_2^2 \right\} \left( \|\boldsymbol{W}_{\boldsymbol{v}}\|_2^2 - \|\boldsymbol{W}^{[2]}\|_2^2 \right) \neq 0. \tag{10}$$

Having clarified the definition of non-degenerate initialization, we present the following theorem that reveals the non-degeneracy property of effective training dynamics.

**Theorem 1** (Blow-up in finite time). *Let the parameters be initialized randomly as above from a Gaussian distribution. Then, almost surely, the initialization is non-degenerate in the sense of Definition 4, and the effective training dynamics Eq. (7) blows up in finite time. That is, there exists $T^* > 0$ such that*

$$\lim_{t \to T^*} E(t) = +\infty.$$

**Proof sketch.** We prove finitetime blow-up via a Riccati-type differential inequality for the energy $E(t)$. Full technical details are provided in Appendix A.1.

*(1) Superlinear growth.* A direct computation gives $\dot{E}(t) \geq 3E(t)^{4/3}$, hence $\partial_t E(t)^{-1/3} \leq -1$ and

$$E(t) \geq \frac{1}{\left( E(0)^{-1/3} - t \right)^3}. \tag{11}$$

For $E(0) > 0$ this yields $T^* \leq E(0)^{-1/3}$.

*(2) Negative initial energy.* If $E(0) \leq 0$, then

$$E(t) \geq -\frac{1}{\left((-E(0))^{-1/3} + t\right)^3},\tag{12}$$

so $E$ is increasing and cannot remain negative indefinitely. Assuming $E(t) \leq 0$ for all $t$ leads to contradictions with (i) standard continuation at finite $T^*$, or (ii) monotone limits at $T^* = \infty$, reducing to the borderline case $E(t) \uparrow 0$.

*(3) Borderline exclusion.* In the regime $T^* = \infty$ and $E(t) \uparrow 0$, structural identities and conservation give $\|\boldsymbol{W_v}(t)\|_2^2 \, \|\boldsymbol{W}^{[2]}(t)\|_2^2 \to 0$ and $\dot{E}(t) \to 0$. Under the non-degenerate initialization (Def. 4), this forces a contradiction, since the limiting $\dot{E}$ must be strictly positive.

## 4.3 Condensation Dynamics

We have proved that energy and parameters norm will blow up almost surely. It implies the effective dynamics drive parameters escape small initialization area in finite time. The next question is how the effective dynamics affects the emergence of condensation and whether there exist observables to help us characterize condensation.

We propose a condition of condensation and verify its effectiveness using experimental and theoretical methods. In particular, we theoretically prove that the solution of the effective dynamics has specific properties, which is to some extent a sufficiency argument. The necessity argument is quite difficult in theory. But experimental results provide us a strong implication that this condition maybe also necessary.

**Assumption 1** (Condensation condition)**.** *The parameters satisfy the **condensation condition** at time $t$. That is*

*1. For each index $i \in [d_m]$, $\boldsymbol{W}_i^{[2]} \boldsymbol{W_v} \boldsymbol{W}^{[1],i} > 0$ and $\boldsymbol{W}_{\boldsymbol{v},i} \boldsymbol{W}_i^{[1]} \boldsymbol{W}^{[2]} > 0$.*

*2. For each pair $i, j \in [d_m]$, $\langle \boldsymbol{W}_i^{[2]} \boldsymbol{W}^{[1],i}, \boldsymbol{W}_j^{[2]} \boldsymbol{W}^{[1],j} \rangle > 0$, and $\langle \boldsymbol{W}_{\boldsymbol{v},i} \boldsymbol{W}_i^{[1]}, \boldsymbol{W}_{\boldsymbol{v},j} \boldsymbol{W}_j^{[1]} \rangle > 0$.*

This hypothesis can be verified experimentally in Sec. 5.1.2. Then, based on Assumption 1, we formalize the statement of the culminating theorem as follows:

**Theorem 2** (Condensation)**.** *Under Assumption 1, the effective dynamical system governed by Eq. (7) drives the parameter matrix $\boldsymbol{W}_V$ to undergo condensation in the sense of Definition 1.*

This section gives a highlevel proof sketch; full details appear in Appendix A.2.

**Proof sketch.** We establish finitetime directional convergence (condensation) via geometric propagation and twosided energy control.

*(1) Geometric consistency and alignment dynamics.* Under Assumption 1, Proposition 4 shows that once the alignment condition holds at some $t_0 < T^*$, it propagates throughout $(t_0, T^*)$. Proposition 5 yields a structural dichotomy of the columns of $\boldsymbol{W}_V$ into a condensing class $C_1$ and a uniformly bounded class $C_2$. Propositions 6 and 7 further establish dynamical alignment between $\boldsymbol{W}^{[2]}$ and its time derivative, ensuring coherence of the evolving direction.

*(2) Singularity structure and condensation.* Proposition 8 supplies an energy upper bound which, combined with the lower bound in Eq. (11), furnishes a bilateral estimate on $E(t)$. A telescoping-integral argument then proves that the condensing indices dominate in finite time, completing the proof of condensation via Theorem 2.

## 4.4 Key-Query Dynamics

Following the initial training stage, the parameter matrices $\boldsymbol{W}_V$, $\boldsymbol{W}^{[1]}$ and $\boldsymbol{W}^{[2]}$ exhibit substantial growth in magnitude, effectively escaping the small-initialization regime. In contrast, the key-query matrices $\boldsymbol{W}_Q$ and $\boldsymbol{W}_K$ demonstrate remarkable stability in scale. This separation phenomenon is fundamentally governed by the effective dynamics of the learning system.

A pivotal question arises: Under what conditions do the key-query matrices become dynamically activated, thereby enabling the attention mechanism to exert its structural influence? We hypothesize that during early training, $\boldsymbol{W}_V$, $\boldsymbol{W}^{[1]}$ and $\boldsymbol{W}^{[2]}$ converge to a critical point where $\boldsymbol{W}_Q$ and $\boldsymbol{W}_K$ almost vanish, temporarily stabilizing in this dormant state. The following analysis provides mechanistic insights into this dynamical freezing phenomenon. The final activation function is omitted from our analysis. This is justified because the layer's pre-activations consistently operate within the linear regime of the function. Furthermore, empirical results confirm that its inclusion does not alter the model's learning dynamics (refer to B.2). Now empirical loss $\mathcal{L}(\boldsymbol{\theta})$ has the following decomposition:

$$\mathcal{L}(\boldsymbol{\theta}) \approx \frac{1}{n} \sum_{i=1}^{n} \mathcal{L}_{1,i}(\boldsymbol{\theta}) \mathcal{L}_{2,i}(\boldsymbol{\theta}), \tag{13}$$

where

$\mathcal{L}_{1,i}(\boldsymbol{\theta}) = \exp\left\{-y_i \left(\sum_{j=1}^{s} \frac{1}{s} \boldsymbol{X}_{i,j} \boldsymbol{W}_V \boldsymbol{W}^{[1]} \boldsymbol{W}^{[2]}\right)\right\}$,

$\mathcal{L}_{2,i}(\boldsymbol{\theta}) = \exp\left\{-y_i \left(\sum_{j=1}^{s} \left(\frac{1}{s} \frac{\boldsymbol{X}_{i,s} \boldsymbol{W}_Q \boldsymbol{W}_K^{\mathsf{T}} \boldsymbol{X}_{i,j}^{\mathsf{T}}}{\sqrt{d_m}} - \frac{1}{s^2} \sum_{l=1}^{s} \frac{\boldsymbol{X}_{i,s} \boldsymbol{W}_Q \boldsymbol{W}_K^{\mathsf{T}} \boldsymbol{X}_{i,l}^{\mathsf{T}}}{\sqrt{d_m}}\right) \boldsymbol{X}_{i,j} \boldsymbol{W}_V \boldsymbol{W}^{[1]} \boldsymbol{W}^{[2]}\right)\right\}$.

Based on the above discussion, we now formalize the following assumption.

**Assumption 2** (Dynamics separation stage). *After the breakdown of the effective dynamics in Eq. (7), let $\delta$ denote a small parameter. The gradient flow subsequently enters a stage characterized by:*

1. **Criticality conditions:** *The outer parameters $(\boldsymbol{W}_V, \boldsymbol{W}^{[1]}, \boldsymbol{W}^{[2]})$ converge to a quasi-stationary configuration such that $\nabla_{\boldsymbol{W}_V} \widetilde{\mathcal{L}}_1 = \nabla_{\boldsymbol{W}^{[1]}} \widetilde{\mathcal{L}}_1 = \nabla_{\boldsymbol{W}^{[2]}} \widetilde{\mathcal{L}}_1 = \mathcal{O}(\delta^2)$, where $\widetilde{\mathcal{L}}_1 = \frac{1}{n} \sum_i \mathcal{L}_{1,i}$.*

2. **Key-query stunting:** *The attention parameters remain small, satisfying $|\boldsymbol{W}_{Q\,ij}|, |\boldsymbol{W}_{K\,ij}| = \mathcal{O}(\delta)$, until their norms $\|\boldsymbol{W}_Q\|$ and $\|\boldsymbol{W}_K\|$ exceed a critical scale.*

To facilitate empirical validation, we introduce a modified version of the basic equivalence of the first part of Assumption 2, denoted as Assumption 2*.

**Assumption 2*.** *The outer parameters $(\boldsymbol{W}_V, \boldsymbol{W}^{[1]}, \boldsymbol{W}^{[2]})$ reach a quasi-stationary state whose directions vary negligibly over time, i.e., $\frac{\mathrm{d}}{\mathrm{d}t}\left(\frac{\boldsymbol{W}_V}{\|\boldsymbol{W}_V\|}\right) = \frac{\mathrm{d}}{\mathrm{d}t}\left(\frac{\boldsymbol{W}^{[1]}}{\|\boldsymbol{W}^{[1]}\|}\right) = \frac{\mathrm{d}}{\mathrm{d}t}\left(\frac{\boldsymbol{W}^{[2]}}{\|\boldsymbol{W}^{[2]}\|}\right) \approx \boldsymbol{0}$, and the loss evolution satisfies $\frac{\mathrm{d}\mathcal{L}_{1,i}}{\mathrm{d}t} = \mathcal{O}(\delta^2)$ for all $i$.*

Since we assume the small parameter $\delta$ is still relatively small, we get the following Proposition which illustrate evident dynamics separation and the leading order dynamics of key-query matrices.

**Proposition 3** (Effective dynamics during dynamics separation stage). *Under Assumption 2 or Assumption 2*, the empirical risk $\mathcal{L}(\boldsymbol{\theta})$ exhibits the following properties:*

1. **Dynamics separation** *The gradients of the empirical risk with respect to $\boldsymbol{W}_V$, $\boldsymbol{W}^{[1]}$ and $\boldsymbol{W}^{[2]}$ are of order $\mathcal{O}(\delta^2)$, while the gradients with respect to the query matrix $\boldsymbol{W}_Q$ and key $\boldsymbol{W}_K$ are of order $\mathcal{O}(\delta)$.*

2. **Key-query dynamics** *Treating $\boldsymbol{W}_V$, $\boldsymbol{W}^{[1]}$ and $\boldsymbol{W}^{[2]}$ as fixed due to dynamics separation, the leading-order dynamics of key-query matrices are given by*

$$\frac{\mathrm{d}\boldsymbol{W}_Q}{\mathrm{d}t} = \boldsymbol{F}\boldsymbol{W}_K, \quad \frac{\mathrm{d}\boldsymbol{W}_K}{\mathrm{d}t} = \boldsymbol{F}^{\mathsf{T}}\boldsymbol{W}_Q, \tag{14}$$

*where $\boldsymbol{F}$ is defined as follows*

$$\boldsymbol{F} = \frac{1}{ns\sqrt{d_m}} \sum_{i=1}^{n} y_i \mathcal{L}_{1,i} \boldsymbol{X}_{i,s}^{\mathsf{T}} \boldsymbol{W}^{[2]\mathsf{T}} \boldsymbol{W}^{[1]\mathsf{T}} \boldsymbol{W}_V^{\mathsf{T}} \left(\sum_{j=1}^{s} \boldsymbol{X}_{i,j}^{\mathsf{T}} \left(\boldsymbol{X}_{i,j} - \frac{1}{s} \sum_{l=1}^{s} \boldsymbol{X}_{i,l}\right)\right). \tag{15}$$

Since the dynamics governing $\boldsymbol{W}_Q$ and $\boldsymbol{W}_K$ form a linear ordinary differential equation system in this context, we can rigorously establish the subsequent conclusions.

**Theorem 3** (Asymptotic rank collapse). *Given the key-query dynamics governed by Eq. (14), the normalized key and query matrices exhibit rank collapse as Definition 2. Specifically, when $\boldsymbol{F}$ possesses a unique largest singular value, both normalized matrices asymptotically become rank 1.*

The detailed proofs of Proposition 3 and Theorem 3 can be found in the Appendix A.3.

## 5 Experimental Results

In this section, we first demonstrate the phenomena of cohesion and rank collapse using synthetic data and confirm the assumptions required for our theoretical analysis of the one-layer Transformer model. We then present experiments on natural language processing tasks to demonstrate the generality of our theoretical findings with respect to various datasets and network architectures.

### 5.1 Synthetic Dataset

We employ the concept of the anchor function Zhang et al. [2024c] to construct a synthetic dataset that simulates a simplified language modeling scenario. The model is a one-layer Transformer with $\tanh$ activation, trained using cross-entropy loss and the AdamW optimizer. Further experimental settings are detailed in Appendix B.1.

#### 5.1.1 Phenomenon: Condensation and Rank Collapse

To dissect the learning dynamics, we visualize the training process through three complementary lenses: the cosine similarity of parameters (Calculation method refer to Sec. B.1), the relative change of norms, and the effective rank of weight matrices. As shown in Figure 1, these analyses collectively reveal a distinct three-stage training trajectory, which we characterize as Condensation, Key-Query Rank Collapse, and the further training.

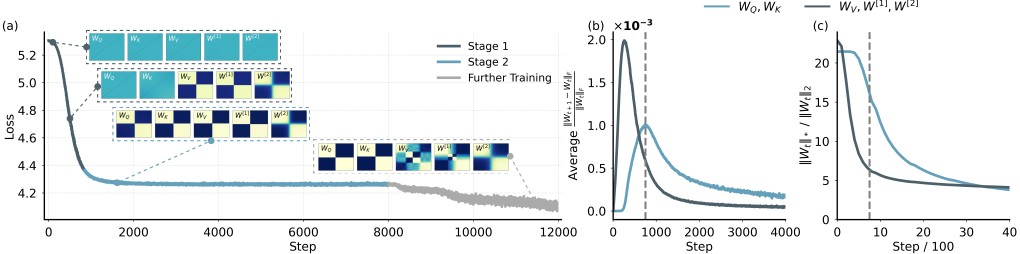

Figure 1: (a) Evolution of cosine similarity matrices for outer and attention parameters. The training process is partitioned into three stages: Condensation (Stage 1), Key-Query rank collapse (Stage 2), and a further training stage. Stage transitions are identified by plateaus in the loss curve and structural shifts in these matrices. (b) The relative change of norms between attention and outer parameters. The gray dashed line marks the onset of Stage 2, where updates to the attention parameters begin to dominate. (c) Evolution of the effective rank for both parameter groups, tracking the change in their intrinsic dimensionality throughout training.

The training process begins with a rapid decrease in loss, driven almost exclusively by the outer-layer parameters since the relative change of the outer parameters far exceed those of the attention parameters (Fig.1(b)) during this initial phase. This intense optimization leads to the condensation phenomenon, where the initially random outer parameters organize into a low-rank configuration. This is visually evident from the emergence of block structures in their cosine similarity matrices (Fig.1(a)) and is quantified by a monotonic and significant decrease in their effective rank (Fig. 1(c)). Throughout this stage, the attention parameters remain largely static and unstructured.

Following the initial phase, the training loss enters a prolonged plateau. This signals a critical transition in the learning dynamics, marked by the gray dashed line in Fig.1(b). At this stage, a clear dynamics separation occurs: the updates to the outer parameters subside, and the attention parameters become the primary focus of optimization. This empirical observation validates our theoretical framework, particularly Proposition 3. As the changes in the outer parameters become slower (supporting Assumption 2), the attention parameters begin to learn their specialized roles. This is characterized by a rank collapse, confirmed visually by the sudden formation of structure in their similarity matrices and quantitatively by a precipitous drop in their effective rank (Fig. 1(a), 1(c)).

### 5.1.2 Experimental Validation of Key Assumptions

To ground our theoretical analysis in the observed dynamics, we now provide direct empirical validation for the key assumptions that underpin our framework: Assumption 1 and Assumption 2.

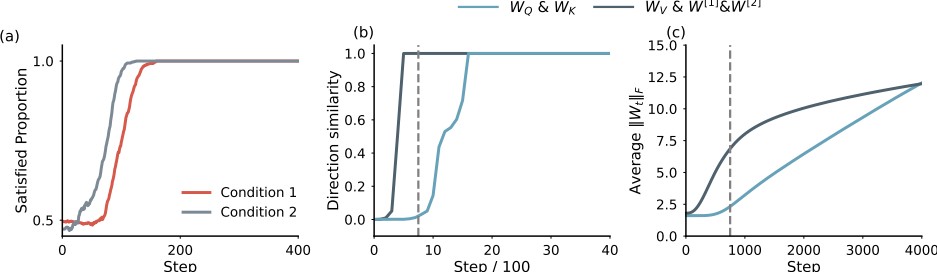

Figure 2: (a) Proportion of satisfied conditions in Assumption 1, measured as $\frac{|A_1|}{d_m}$ and $\frac{|A_2|}{d_m^2}$ (Definitions of $A_1$ and $A_2$ refer to Sec. B.1). (b) Similarity between singular vectors of two adjacent time steps. For example, let $U_t \Sigma_t V_t$ and $U_{t+1} \Sigma_{t+1} V_{t+1}$ be the singular value decompositions of parameter matrix $W_t$ and $W_{t+1}$. The similarity is defined as $\frac{1}{d_m} \sum_{i=1}^{d_m} \cos(u_t^i, u_{t+1}^i)$ (or $\frac{1}{d_m} \sum_{i=1}^{d_m} \cos(v_t^i, v_{t+1}^i)$). (c) Frobenius norms of parameter groups.

First, we examine the condensation condition. Figure 2(a) plots the proportion of satisfied conditions in Assumption 1. The proportion rapidly approaches 1 within the first 200 training steps, confirming that the outer parameters quickly converge to a state where this assumption holds.

Next, we validate the assumption of dynamics separation. As discussed in the previous section, our observation that the gradual change of outer parameters during Stage 2 and flat loss curve (often means a critical point has appeared) already provide strong qualitative support for the first part of Assumption 2. To analyze this more rigorously, we examine its empirical variant, Assumption 2*. This assumption points that the direction of parameters remains unchanged and the leading-order loss changes very slowly.

Figure 2(b) shows the cosine similarity between the singular vectors of the outer parameter matrices at adjacent time steps. The similarity for all outer parameters remains extremely close to 1 after the first stage. This indicates that the subspace spanned by these parameters is highly stable, meaning their directional structure is effectively frozen. This stability, combined with the flat loss curve observed in Stage 2, provides compelling evidence for Assumption 2*. Figure 2(c) validates the scale separation implied by the assumption. It shows that by the onset of Stage 2, the Frobenius norms of the outer parameters have grown significantly, while the norms of the attention parameters remain small and close to their initialization values. This confirms the expected scale difference between the two parameter groups, where outer parameters are $\mathcal{O}(1)$ and attention parameters are $\mathcal{O}(\delta)$.

### 5.2 Real Task

We further validate our theoretical predictions on a real-world language modeling benchmark, Wiki-Text Merity et al. [2017]. Unlike the synthetic setup, where anchor functions are explicitly defined, WikiText provides natural linguistic dependencies and high distributional variability. This allows us to test whether the proposed two-stage dynamics, early condensation of outer parameters followed by attention-driven rank collapse, persist in realistic Transformer training. In this setting, we employ a two-layer transformer with GeLU activation and residual connections. To keep the consistency of architecture and focused on the core dynamics, layer normalization is omitted. Further experimental settings are provided in Appendix B.3.

As shown in Figure 3, the two-layer Transformer on WikiText exhibits the same stage-wise dynamics observed in the synthetic experiments. During the initial phase, the outer parameters $(W_V, W^{[1]}, W^{[2]})$ in both layers undergo rapid condensation, while the attention weights

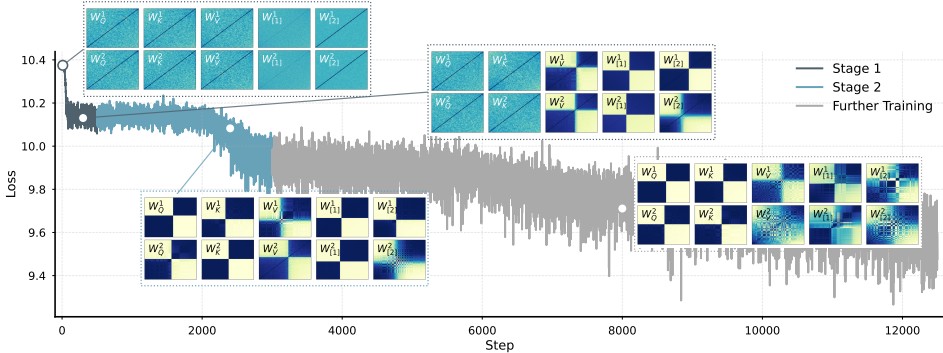

Figure 3: Evolution of cosine similarity between parameter of the two-layer transformer on WikiText dataset. Training dynamics also show a similar three-phase characteristic. Superscripts are used to indicate parameters of different layers, and subscripts indicate different parameters within a layer. For example, $\boldsymbol{W}_V^1$ represents the value matrix of the first layer.

$(\boldsymbol{W}_Q, \boldsymbol{W}_K)$ remain largely unchanged. As training proceeds and the loss enters a plateau, the attention parameters begin to evolve, displaying a sharp rank collapse that reorganizes internal representations.

This empirical observation confirms that the separation between outer-parameter condensation and attention-driven rank reduction is not an artifact of the synthetic dataset but also emerges naturally in real-world text modeling. The consistent appearance of this two-stage dynamic across both synthetic and natural settings suggests that implicit regularization, first through low-rank condensation and then through targeted attention adaptation, may serve as a general mechanism underlying the emergence of structured representations in Transformer models.

# 6 Discussion

## 6.1 Conclusion

This work advances the theoretical understanding of transformer training dynamics by establishing a two-stage analytical framework. Through gradient flow analysis, we show that small initialization helps models escape degenerate regions via asymmetric weight updates, leading to condensation of parameter matrices toward task-relevant directions. In the subsequent stage, the key-query matrices undergo a coordinated collapse that further refines the learned representations. Together, these results clarify the mechanisms underlying the condensation and rank collapse phenomena, providing a principled foundation for future studies on Transformer optimization and generalization.

## 6.2 Limitations

While this work provides valuable theoretical insights, its most significant constraint stems from analyzing exclusively binary classification scenarios: a simplification dictated by technical barriers in gradient flow analysis. This narrow scope inherently precludes insights into transformers' dynamics in practical multi-class classification or sequence-to-sequence learning contexts, where complex interactions between multiple prediction targets and attention mechanisms likely emerge. Though focused theoretical simplification is methodologically justified, extending this framework to broader problem domains remains critical for unifying theory with real-world transformer optimization. Future work should prioritize overcoming these technical limitations to theoretically verify whether our conclusions hold true beyond binary settings.

## Acknowledgments and Disclosure of Funding

This work is sponsored by the National Key R&D Program of China Grant No. 2022YFA1008200 (T. L.). We also thank Shanghai Institute for Mathematics and Interdisciplinary Sciences (SIMIS) for their financial support. This research was funded by SIMIS under grant number SIMIS-ID-2025-ST. The authors are grateful for the resources and facilities provided by SIMIS, which were essential for the completion of this work. We thank Pengxiao Lin for insightful discussions and support and encouragement to the authors.

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

## A Theory Details

### A.1 Theory Details for Blow up Dynamics

#### A.1.1 Proof for Proposition 2

*Proof.* Taking advantage of the inherent symmetry of the system, the proof focuses on analyzing the coupled dynamics of $\boldsymbol{W_v}$ and $\boldsymbol{W}^{[1]}$:

$$\frac{\mathrm{d}}{\mathrm{d}t}\left(\boldsymbol{W_{v,k}}\right)^2 = 2\sum_{k'}\boldsymbol{W_{v,k}}\boldsymbol{W}^{[1]}_{kk'}\boldsymbol{W}^{[2]}_{k'} = \frac{\mathrm{d}}{\mathrm{d}t}\sum_{k'}\left(\boldsymbol{W}^{[1]}_{kk'}\right)^2.$$

This finished the proof of the first two equations. We also derive the relation between energy $E$ and the evolution of parameters

$$\frac{\mathrm{d}}{\mathrm{d}t}\|\boldsymbol{W_v}\|_2^2 = \frac{\mathrm{d}}{\mathrm{d}t}\sum_k \boldsymbol{W}^2_{v,k} = 2\sum_{k,k'}\boldsymbol{W_{v,k}}\boldsymbol{W}^{[1]}_{kk'}\boldsymbol{W}^{[2]}_{k'} = 2E.$$

This is just the third equation. $\qquad\square$

#### A.1.2 Proof for Theorem 1

*Proof.* Since we use Gaussian random initialization, the initialization satisfies Definition 4 almost surely. Therefore, we establish our results under the assumptions specified in Definition 4. By local Lipshcitz condition on the right hand side of dynamical system Eq. (7), it has a solution for $t \in (0, T^*)$ where $T^*$ is maximum existence time of solution and can be infinity. Taking derivative of $E$, we obtain

$$\dot{E} = \frac{\mathrm{d}}{\mathrm{d}t}\boldsymbol{W_v}\boldsymbol{W}^{[1]}\boldsymbol{W}^{[2]} = \|\dot{\boldsymbol{W}}_v\|_2^2 + \|\dot{\boldsymbol{W}}^{[2]}\|_2^2 + \|\boldsymbol{W_v}\|_2^2\|\boldsymbol{W}^{[2]}\|_2^2. \tag{16}$$

The inequality of arithmetic and geometric means leads to

$$\begin{aligned}\dot{E} &\geq 3(\|\dot{\boldsymbol{W}}_v\|_2^2\|\dot{\boldsymbol{W}}^{[2]}\|_2^2\|\boldsymbol{W_v}\|_2^2\|\boldsymbol{W}^{[2]}\|_2^2)^{\frac{1}{3}} \\ &\geq [\langle\dot{\boldsymbol{W}}_v\boldsymbol{W_v}\rangle^2\langle\dot{\boldsymbol{W}}^{[2]}\boldsymbol{W}^{[2]}\rangle^2]^{\frac{1}{3}} \\ &= 3E^{\frac{4}{3}}.\end{aligned}$$

This implies that energy $E$ increase monotonically. If $E(0) > 0$,

$$\frac{\mathrm{d}}{\mathrm{d}t}E^{-\frac{1}{3}} \leq -1.$$

Integrating both sides of the inequality yields a lower bound for the energy $E$

$$E(t) \geq \frac{1}{(E(0)^{-\frac{1}{3}} - t)^3}. \tag{17}$$

Thus, in the case where $E(0) > 0$, the dynamical system explodes before $T^* \leq E(0)^{-\frac{1}{3}}$.

In the case where $E(0) \leq 0$, we consider $-E(t)$ instead, and obtain

$$E(t) \geq -\frac{1}{(t + (-E(0))^{-\frac{1}{3}})^3}. \tag{18}$$

We claim that there exists some time $t_0 > 0$, such that $E(t_0) > 0$. This claim can be proved by contradiction.

Suppose that $E(t) \leq 0$ for all $0 < t < T^*$. Recall that $\dot{E}(t) \geq 0$ throughout this interval. The boundedness of $\boldsymbol{W_v}$, $\boldsymbol{W}^{[1]}$, and $\boldsymbol{W}^{[2]}$, together with the monotonicity of energy $E$, implies that $E(T^*) = \lim_{t \to T^*} E(t)$ exists and satisfies $-\infty < E(T^*) \leq 0$. We now consider different cases separately.

(i) The case of $T^* < +\infty$. The solutions can be extended to a time larger than $T^*$ since $\|\boldsymbol{W_v}\|_2$, $\|\boldsymbol{W}^{[1]}\|_F$, $\|\boldsymbol{W}^{[2]}\|_2$ are bounded due to the conservation law. This contradicts the definition of $T^*$.

(ii) The case of $T^* = +\infty$ and $E(T^*) < 0$. That is $\lim_{t \to +\infty} E(t) < 0$. However, this contradicts Eq. (18).

(iii) The case of $T^* = +\infty$ and $E(T^*) = 0$. That is $\lim_{t \to +\infty} E(t) = 0$. We prove this case in three steps.

**Step 1**: We show that $\lim_{t \to \infty} \|\boldsymbol{W_v}(t)\|_2^2 \|\boldsymbol{W}^{[2]}(t)\|_2^2 = 0$. Since $E(t) \leq 0$ for all $t$, the quantities $\|\boldsymbol{W_v}\|_2^2$, $\|\boldsymbol{W}^{[1]}\|_{\mathrm{F}}^2$, $\|\boldsymbol{W}^{[2]}\|_2^2$ are monotonically decreasing. However, each of them is bounded below by zero, and hence they all converge to finite limits and remain uniformly bounded.

Moreover, note that $\dot{E} \geq 0$ for all $t$. If $\liminf_{t \to \infty} \dot{E}(t) > 0$, it contradicts the fact that $\lim_{t \to \infty} E(t) = 0$. Therefore, it must hold that

$$\lim_{t \to \infty} \|\boldsymbol{W_v}(t)\|_2^2 \|\boldsymbol{W}^{[2]}(t)\|_2^2 = 0.$$

This implies that either $\lim_{t \to \infty} \|\boldsymbol{W_v}(t)\|_2^2 = 0$ or $\lim_{t \to \infty} \|\boldsymbol{W}^{[2]}(t)\|_2^2 = 0$. We can also obtain $\lim_{t \to \infty} \|\dot{\boldsymbol{W}}^{[1]}(t)\|_{\mathrm{F}}^2 = 0$ since $\dot{\boldsymbol{W}}^{[1]} = \boldsymbol{W_v}^{\mathsf{T}} \boldsymbol{W}^{[2]}$.

**Step 2**: We show $\lim_{t \to \infty} \dot{E}(t) = 0$. Without loss of generality, we assume that $\lim_{t \to \infty} \|\boldsymbol{W}^{[2]}(t)\|_2^2 = 0$. The case of $\lim_{t \to \infty} \|\boldsymbol{W_v}(t)\|_2^2 = 0$ is similar. That is $\|\boldsymbol{W_v}(0)\|_2 > \|\boldsymbol{W}^{[2]}\|_2$. By conservation law, we have $\lim_{t \to \infty} \|\dot{\boldsymbol{W}}_{\boldsymbol{v}}(t)\|_2^2 = 0$.

Considering the second derivative of $\boldsymbol{W}_{k'}^{[2]}$, we obtain

$$\ddot{\boldsymbol{W}}_{k'}^{[2]} = \sum_k \left( \sum_l \boldsymbol{W}_{kl}^{[1]} \boldsymbol{W}_l^{[2]} \boldsymbol{W}_{kk'}^{[1]} + \boldsymbol{W}_{\boldsymbol{v},k}^2 \boldsymbol{W}_{k'}^{[1]} \right).$$

Thus $\lim_{t \to \infty} \|\ddot{\boldsymbol{W}}^{[1]}(t)\|_2^2 = 0$ since $\lim_{t \to \infty} \|\boldsymbol{W}^{[1]}(t)\|_2^2 = 0$. Note that $\dot{E}(t) \geq 0$ and $\lim_{t \to \infty} E(t) = 0$. Recall that $\liminf_{t \to \infty} \dot{E}(t) = 0$ and $\dot{E}$ are bounded. And we also have $\|\dot{\boldsymbol{W}}^{[2]}\|_2^2 \leq M$. We claim that $\lim_{t \to \infty} \|\dot{\boldsymbol{W}}^{[2]}(t)\|_2^2 = 0$, which implies

$$\lim_{t \to \infty} \dot{E}(t) = 0. \tag{19}$$

Since $\lim_{t \to \infty} \|\boldsymbol{W}^{[2]}(t)\|_2 = 0$ and $\lim_{t \to \infty} \|\ddot{\boldsymbol{W}}^{[2]}(t)\|_2 = 0$. Using Taylor expansion, we have for some $\varphi \in [t, t+1]$

$$\boldsymbol{W}_k^{[2]}(t+1) = \boldsymbol{W}_k^{[2]}(t) + \dot{\boldsymbol{W}}_k^{[2]}(t) + \frac{1}{2} \ddot{\boldsymbol{W}}_k^{[2]}(\varphi), \forall k,$$

which implies $\lim_{t \to \infty} \|\dot{\boldsymbol{W}}_{\boldsymbol{v}}(t)\|_2 = 0$. Therefore, the assertion holds.

**Step 3**: We show that Eq. (19) contradicts the condition in Definition 4. By direct calculation, we obtain

$$\begin{aligned}
\frac{\mathrm{d}}{\mathrm{d}t} \|\dot{\boldsymbol{W}}_{\boldsymbol{v}}\|_2^2 &= \frac{\mathrm{d}}{\mathrm{d}t} (\boldsymbol{W}^{[2]\mathsf{T}} \boldsymbol{W}^{[1]\mathsf{T}} \boldsymbol{W}^{[1]} \boldsymbol{W}^{[2]}) \\
&= \dot{\boldsymbol{W}}^{[2]\mathsf{T}} \boldsymbol{W}^{[1]\mathsf{T}} \boldsymbol{W}^{[1]} \boldsymbol{W}^{[2]} + \boldsymbol{W}^{[2]\mathsf{T}} \dot{\boldsymbol{W}}^{[1]\mathsf{T}} \boldsymbol{W}^{[1]} \boldsymbol{W}^{[2]} \\
&\quad + \boldsymbol{W}^{[2]\mathsf{T}} \boldsymbol{W}^{[1]\mathsf{T}} \dot{\boldsymbol{W}}^{[1]} \boldsymbol{W}^{[2]} + \boldsymbol{W}^{[2]\mathsf{T}} \boldsymbol{W}^{[1]\mathsf{T}} \boldsymbol{W}^{[1]} \dot{\boldsymbol{W}}^{[2]} \\
&= \boldsymbol{W_v} \boldsymbol{W}^{[1]} \boldsymbol{W}^{[1]\mathsf{T}} \boldsymbol{W}^{[1]} \boldsymbol{W}^{[2]} + \boldsymbol{W}^{[2]\mathsf{T}} \boldsymbol{W}^{[2]} \boldsymbol{W_v} \boldsymbol{W}^{[1]} \boldsymbol{W}^{[2]} \\
&\quad + \boldsymbol{W}^{[2]\mathsf{T}} \boldsymbol{W}^{[1]\mathsf{T}} \boldsymbol{W_v}^{\mathsf{T}} \boldsymbol{W}^{[2]\mathsf{T}} \boldsymbol{W}^{[2]} + \boldsymbol{W}^{[2]\mathsf{T}} \boldsymbol{W}^{[1]\mathsf{T}} \boldsymbol{W}^{[1]} \boldsymbol{W}^{[1]\mathsf{T}} \boldsymbol{W_v}^{\mathsf{T}} \\
&= 2E \|\boldsymbol{W}^{[1]}\|_2^2 + 2 \dot{\boldsymbol{W}}_{\boldsymbol{v}} \boldsymbol{W}^{[1]} \dot{\boldsymbol{W}}^{[2]}
\end{aligned}$$

and

$$\begin{aligned}
\frac{\mathrm{d}}{\mathrm{d}t} \|\dot{\boldsymbol{W}}^{[2]}\|_2^2 &= \frac{\mathrm{d}}{\mathrm{d}t} (\boldsymbol{W_v} \boldsymbol{W}^{[1]} \boldsymbol{W}^{[1]\mathsf{T}} \boldsymbol{W_v}^{\mathsf{T}}) \\
&= \dot{\boldsymbol{W}}_{\boldsymbol{v}} \boldsymbol{W}^{[1]} \boldsymbol{W}^{[1]\mathsf{T}} \boldsymbol{W_v}^{\mathsf{T}} + \boldsymbol{W_v} \dot{\boldsymbol{W}}^{[1]} \boldsymbol{W}^{[1]\mathsf{T}} \boldsymbol{W_v}^{\mathsf{T}} \\
&\quad + \boldsymbol{W_v} \boldsymbol{W}^{[1]} \dot{\boldsymbol{W}}^{[1]\mathsf{T}} \boldsymbol{W_v}^{\mathsf{T}} + \boldsymbol{W_v} \boldsymbol{W}^{[1]} \boldsymbol{W}^{[1]\mathsf{T}} \dot{\boldsymbol{W}}_{\boldsymbol{v}}^{\mathsf{T}} \\
&= \boldsymbol{W}^{[2]\mathsf{T}} \boldsymbol{W}^{[1]\mathsf{T}} \boldsymbol{W}^{[1]} \boldsymbol{W}^{[1]\mathsf{T}} \boldsymbol{W_v}^{\mathsf{T}} + \boldsymbol{W_v} \boldsymbol{W_v}^{\mathsf{T}} \boldsymbol{W}^{[2]\mathsf{T}} \boldsymbol{W}^{[1]\mathsf{T}} \boldsymbol{W_v}^{\mathsf{T}} \\
&\quad + \boldsymbol{W_v} \boldsymbol{W}^{[1]} \boldsymbol{W}^{[2]} \boldsymbol{W_v} \boldsymbol{W_v}^{\mathsf{T}} + \boldsymbol{W_v} \boldsymbol{W}^{[1]} \boldsymbol{W}^{[1]\mathsf{T}} \boldsymbol{W}^{[1]} \boldsymbol{W}^{[2]} \\
&= 2E \|\boldsymbol{W_v}\|_2^2 + 2 \dot{\boldsymbol{W}}_{\boldsymbol{v}} \boldsymbol{W}^{[1]} \dot{\boldsymbol{W}}^{[2]}.
\end{aligned}$$

Therefore,

$$\frac{\mathrm{d}}{\mathrm{d}t}\|\dot{\boldsymbol{W}}_{\boldsymbol{v}}\|_2^2 - \frac{\mathrm{d}}{\mathrm{d}t}\|\dot{\boldsymbol{W}}^{[2]}\|_2^2 = 2E(\|\boldsymbol{W}^{[2]}(0)\|_2^2 - \|\boldsymbol{W}_{\boldsymbol{v}}(0)\|_2^2).$$

Integrating both sides of the equality, we obtain

$$\lim_{t\to\infty}\|\dot{\boldsymbol{W}}_{\boldsymbol{v}}(t)\|_2^2 - \|\dot{\boldsymbol{W}}^{[2]}(t)\|_2^2 = \|\dot{\boldsymbol{W}}_{\boldsymbol{v}}(0)\|_2^2 - \|\dot{\boldsymbol{W}}^{[2]}(0)\|_2^2 - \|\boldsymbol{W}^{[2]}(0)\|_2^2(\|\boldsymbol{W}^{[2]}(0)\|_2^2 - \|\boldsymbol{W}_{\boldsymbol{v}}(0)\|_2^2).$$

However, according to Definition 4 and the fact that $\lim_{t\to\infty}\|\dot{\boldsymbol{W}}^{[2]}(t)\|_2 = 0$, we have that

$$\lim_{t\to\infty}\|\dot{\boldsymbol{W}}_{\boldsymbol{v}}(t)\|_2^2 \neq 0.$$

Based on Eq. (16), we have

$$\lim_{t\to\infty}\dot{E}(t) = \lim_{t\to\infty}\|\dot{\boldsymbol{W}}_{\boldsymbol{v}}(t)\|_2^2 + \|\dot{\boldsymbol{W}}^{[2]}(t)\|_2^2 + \|\boldsymbol{W}_{\boldsymbol{v}}(t)\|_2^2\|\boldsymbol{W}^{[2]}(t)\|_2^2$$

$$= \lim_{t\to\infty}\|\dot{\boldsymbol{W}}_{\boldsymbol{v}}(t)\|_2^2 \neq 0.$$

It contradicts with Eq. (19) which claims $\lim_{t\to\infty}\dot{E}(t) = 0$. This completes the proof. $\qquad\square$

## A.2 Theory Details for Condensation

In this section, we prove the main theorems which characterize the condensation. In retrospect of the proof of Theorem 1, Eq. (17) provides a lower bound that leads to the presence of explosion. However, this inequality leaves the precise growth rate of energy $E$ undetermined. The key idea here is that Assumption 1 can give us an upper limit on how fast the energy can grow. Once we understand this growth rate, we can then move forward with proving the main theorems.

We begin our proof by the following proposition.

**Proposition 4** (induction). *Consider dynamical system Eq. (7). If Assumption 1 holds at some time $t_0$ with $t_0 < T^*$, then Assumption 1 will hold at $t \in (t_0, T^*)$.*

*Proof.* First, we consider the second condition in Assumption 1. By direct calculation, we have

$$\frac{\mathrm{d}}{\mathrm{d}t}\left\langle \boldsymbol{W}_i^{[2]}\boldsymbol{W}^{[1],i}, \boldsymbol{W}_j^{[2]}\boldsymbol{W}^{[1],j}\right\rangle = \left(\boldsymbol{W}_j^{[2]}\boldsymbol{W}_{\boldsymbol{v}}\boldsymbol{W}^{[1],j}\right)\left(\left(\boldsymbol{W}_i^{[2]}\right)^2 + \frac{1}{\left(\boldsymbol{W}_j^{[2]}\right)^2}\boldsymbol{W}_i^{[2]}\boldsymbol{W}_j^{[2]}\boldsymbol{W}^{[1],i\mathsf{T}}\boldsymbol{W}^{[1],j}\right)$$

$$+ \left(\boldsymbol{W}_i^{[2]}\boldsymbol{W}_{\boldsymbol{v}}\boldsymbol{W}^{[1],i}\right)\left(\left(\boldsymbol{W}_j^{[2]}\right)^2 + \frac{1}{\left(\boldsymbol{W}_i^{[2]}\right)^2}\boldsymbol{W}_i^{[2]}\boldsymbol{W}_j^{[2]}\boldsymbol{W}^{[1],i\mathsf{T}}\boldsymbol{W}^{[1],j}\right),$$

and

$$\frac{\mathrm{d}}{\mathrm{d}t}\left\langle \boldsymbol{W}_{\boldsymbol{v},i}\boldsymbol{W}_i^{[1]}, \boldsymbol{W}_{\boldsymbol{v},j}\boldsymbol{W}_j^{[1]}\right\rangle = \left(\boldsymbol{W}_{\boldsymbol{v},j}\boldsymbol{W}_j^{[1]}\boldsymbol{W}^{[2]}\right)\left(\boldsymbol{W}_{\boldsymbol{v},i}^2 + \frac{1}{\boldsymbol{W}_{\boldsymbol{v},j}^2}\boldsymbol{W}_{\boldsymbol{v},i}\boldsymbol{W}_{\boldsymbol{v},j}\boldsymbol{W}_i^{[1]}\boldsymbol{W}_j^{[1]\mathsf{T}}\right)$$

$$+ \left(\boldsymbol{W}_{\boldsymbol{v},i}\boldsymbol{W}_i^{[1]}\boldsymbol{W}^{[2]}\right)\left(\boldsymbol{W}_{\boldsymbol{v},j}^2 + \frac{1}{\boldsymbol{W}_{\boldsymbol{v},i}^2}\boldsymbol{W}_{\boldsymbol{v},i}\boldsymbol{W}_{\boldsymbol{v},j}\boldsymbol{W}_i^{[1]}\boldsymbol{W}_j^{[1]\mathsf{T}}\right).$$

By Assumption 1, we know the above equations are larger than 0. So $\left\langle \boldsymbol{W}_i^{[2]}\boldsymbol{W}^{[1],i}, \boldsymbol{W}_j^{[2]}\boldsymbol{W}^{[1],j}\right\rangle$ and $\left\langle \boldsymbol{W}_{\boldsymbol{v},i}\boldsymbol{W}_i^{[1]}, \boldsymbol{W}_{\boldsymbol{v},j}\boldsymbol{W}_j^{[1]}\right\rangle$ will be monotonically increasing since $t_0$.

Calculating the derivative of left hand side of first condition, we have

$$\frac{\mathrm{d}}{\mathrm{d}t}\boldsymbol{W}_i^{[2]}\boldsymbol{W}_{\boldsymbol{v}}\boldsymbol{W}^{[1],i} = \dot{\boldsymbol{W}}_{\boldsymbol{v},i}^2 + \sum_{j=1}^{d_m}\boldsymbol{W}_{\boldsymbol{v},i}\boldsymbol{W}_{\boldsymbol{v},j}\boldsymbol{W}^{[1],j\mathsf{T}}\boldsymbol{W}^{[1],i} + \left(\boldsymbol{W}_i^{[2]}\right)^2\|\boldsymbol{W}_{\boldsymbol{v}}\|_2^2$$

and

$$\frac{\mathrm{d}}{\mathrm{d}t}\boldsymbol{W}_{\boldsymbol{v},i}\boldsymbol{W}_i^{[1]}\boldsymbol{W}^{[2]} = \dot{\boldsymbol{W}}_{\boldsymbol{v},i}^2 + \sum_{j=1}^{d_m}\boldsymbol{W}_{\boldsymbol{v},i}\boldsymbol{W}_{\boldsymbol{v},j}\boldsymbol{W}_i^{[1]}\boldsymbol{W}_j^{[1]\mathsf{T}} + (\boldsymbol{W}_{\boldsymbol{v},i})^2\|\boldsymbol{W}^{[2]}\|_2^2.$$

Hence, $\boldsymbol{W}_i^{[2]}\boldsymbol{W_v}\boldsymbol{W}^{[1],i}$ and $\boldsymbol{W}_{\boldsymbol{v},i}\boldsymbol{W}_i^{[1]}\boldsymbol{W}^{[2]}$ will also increase monotonically since $t_0$. Therefore the condensation condition will hold until $T^*$. $\qquad\square$

Next, we analyze the angle relation between $\boldsymbol{W}^{[2]}$ and its derivative $\dot{\boldsymbol{W}}^{[2]}$. For the simplicity of proof and description, we adopt a standardized notation to represent angles between distinct vectors throughout the ensuing discussion.

**Definition 5.** *Let $\xi_{ij}(t)$ denote the angle between the vectors $\boldsymbol{W}_{\boldsymbol{v},i}(t)\boldsymbol{W}_i^{[1]}(t)$ and $\boldsymbol{W}_{\boldsymbol{v},j}(t)\boldsymbol{W}_i^{[1]}(t)$, and $\psi_i(t)$ denote the angle between the vectors $\dot{\boldsymbol{W}}^{[2]}(t)$ and $\boldsymbol{W}_{\boldsymbol{v},i}(t)\boldsymbol{W}_i^{[1]}(t)$. Let $\varphi_i(t)$ denote the angle between $\boldsymbol{W}^{[2]}(t)$ and $\boldsymbol{W}_{\boldsymbol{v},i}(t)\boldsymbol{W}_i^{[1]}(t)$, while $\zeta(t)$ denote the angle between $\boldsymbol{W}^{[2]}(t)$ and $\dot{\boldsymbol{W}}^{[2]}(t)$. In subsequent expressions, the variable $t$ will be omitted unless there is a specific emphasis on the temporal change of angles.*

We divide the entries of vector $\boldsymbol{W_v}$ into two classes according to whether their limit is finite.

**Proposition 5.** *Suppose that Assumption 1 holds. Consider the effective dynamics Eq. (7). The indices $[d_m]$ can be partitioned into two disjoint classes, denoted by $C_1 = \{i_1, \ldots, i_k\} \neq \varnothing$ and $C_2 = [d_m] \setminus C_1$. The partition satisfies the following properties:*

*(i) For each $i \in [m]$, the limits of $\boldsymbol{W}_{\boldsymbol{v},i}$ exist. In particular,*

$$\lim_{t \to T^*} \boldsymbol{W}_{\boldsymbol{v},i} = \begin{cases} \pm \infty, & i \in C_1, \\ \boldsymbol{W}_{\boldsymbol{v},i}^*, & i \in C_2. \end{cases} \tag{20}$$

*(ii) The angle $\xi_{ij}$ between the vectors $\boldsymbol{W}_{\boldsymbol{v},i}(t)\boldsymbol{W}_i^{[1]}(t)$ and $\boldsymbol{W}_{\boldsymbol{v},j}(t)\boldsymbol{W}_i^{[1]}(t)$, as defined in Definition 5, fulfills the condition:*

$$\lim_{t \to T^*} \cos \xi_{ij} = 1, \quad \text{for } i, j \in C_1. \tag{21}$$

*(iii) The following limits exist*

$$\lim_{t \to T^*} \frac{\|\dot{\boldsymbol{W}}^{[2]}\|_2}{\|\boldsymbol{W_v}\|_2^2} = \lim_{t \to T^*} \frac{\|\dot{\boldsymbol{W}}^{[2]}\|_2}{\|\boldsymbol{W}^{[2]}\|_2^2} = 1. \tag{22}$$

*Proof.* 1. First, we find that for every index $i$ the $(\boldsymbol{W}_{\boldsymbol{v},i})^2$ increases monotonically. So their limits exist. We define the index set of the parameters that tend to infinity as $C_1$ and the others as $C_2$. Based on Theorem 1 and conservation laws, we know that $C_1 \neq \varnothing$. Property 1 is automatically satisfied due to our partition.

2. We introduce new variables

$$\begin{cases} p = \langle \boldsymbol{W}_{\boldsymbol{v},i}\boldsymbol{W}_i^{[1]}, \boldsymbol{W}_{\boldsymbol{v},j}\boldsymbol{W}_j^{[1]} \rangle, \\ q = \boldsymbol{W}_{\boldsymbol{v},i}^2 \boldsymbol{W}_{\boldsymbol{v},j}^2. \end{cases}$$

According to the proof of Proposition 4, we find that

$$\begin{aligned} \frac{\mathrm{d}p}{\mathrm{d}t} &= \left(\boldsymbol{W}_{\boldsymbol{v},j}\boldsymbol{W}_j^{[1]}\boldsymbol{W}^{[2]}\right)\left(\boldsymbol{W}_{\boldsymbol{v},i}^2 + \frac{1}{\boldsymbol{W}_{\boldsymbol{v},j}^2}\boldsymbol{W}_{\boldsymbol{v},i}\boldsymbol{W}_{\boldsymbol{v},j}\boldsymbol{W}_i^{[1]}\boldsymbol{W}_j^{[1]\mathsf{T}}\right) \\ &\quad + \left(\boldsymbol{W}_{\boldsymbol{v},i}\boldsymbol{W}_i^{[1]}\boldsymbol{W}^{[2]}\right)\left(\boldsymbol{W}_{\boldsymbol{v},j}^2 + \frac{1}{\boldsymbol{W}_{\boldsymbol{v},i}^2}\boldsymbol{W}_{\boldsymbol{v},i}\boldsymbol{W}_{\boldsymbol{v},j}\boldsymbol{W}_i^{[1]}\boldsymbol{W}_j^{[1]\mathsf{T}}\right) \\ &= \left(\boldsymbol{W}_{\boldsymbol{v},j}\boldsymbol{W}_j^{[1]}\boldsymbol{W}^{[2]}\boldsymbol{W}_{\boldsymbol{v},i}^2 + \boldsymbol{W}_{\boldsymbol{v},i}\boldsymbol{W}_i^{[1]}\boldsymbol{W}^{[2]}\boldsymbol{W}_{\boldsymbol{v},j}^2\right)\left(1 + \frac{p}{q}\right). \end{aligned} \tag{23}$$

Thanks to Eq. (7), we obtain

$$\frac{\mathrm{d}q}{\mathrm{d}t} = 2\left(\boldsymbol{W}_{\boldsymbol{v},j}\boldsymbol{W}_j^{[1]}\boldsymbol{W}^{[2]}\boldsymbol{W}_{\boldsymbol{v},i}^2 + \boldsymbol{W}_{\boldsymbol{v},i}\boldsymbol{W}_i^{[1]}\boldsymbol{W}^{[2]}\boldsymbol{W}_{\boldsymbol{v},j}^2\right). \tag{24}$$

Combining Eq. (23) and Eq. (24), we obtain

$$\frac{\mathrm{d}p}{\mathrm{d}q} = \frac{1}{2}\frac{p}{q} + \frac{1}{2}. \tag{25}$$

Let $u = p/q$. Note that $\frac{\mathrm{d}p}{\mathrm{d}q} = \frac{\mathrm{d}}{\mathrm{d}q}(uq) = q\frac{\mathrm{d}u}{\mathrm{d}q} + u$. Combining this with the right hand of Eq. (25), we get

$$\frac{\mathrm{d}q}{q} = \frac{2\mathrm{d}u}{1 - u}. \tag{26}$$

The Eq. (26) can be solved explicitly.

$$\ln|q(t)| - \ln|q(t_0)| = -2\ln|u(t) - 1| + 2\ln|u(t_0) - 1|. \tag{27}$$

For $i, j \in C_1$, we have $\lim_{t \to T^*} u(t) = 1$ since $q$ tends to infinite as $t$ tends to $T^*$.

By definition,

$$u = \frac{p}{q} = \frac{\langle \boldsymbol{W}_{\boldsymbol{v},i}\boldsymbol{W}_i^{[1]}, \boldsymbol{W}_{\boldsymbol{v},j}\boldsymbol{W}_j^{[1]}\rangle}{\boldsymbol{W}_{\boldsymbol{v},i}^2\boldsymbol{W}_{\boldsymbol{v},j}^2} = \frac{\|\boldsymbol{W}_i^{[1]}\|_2\|\boldsymbol{W}_j^{[1]}\|_2}{|\boldsymbol{W}_{\boldsymbol{v},i}||\boldsymbol{W}_{\boldsymbol{v},j}|}\cos\xi_{ij}. \tag{28}$$

Using the conservation laws, we have

$$\|\boldsymbol{W}_i^{[1]}\|_2^2(t) - \|\boldsymbol{W}_i^{[1]}\|_2^2(0) = \boldsymbol{W}_{\boldsymbol{v},i}^2(t) - \boldsymbol{W}_{\boldsymbol{v},i}^2(0).$$

For $i \in C_1$, we have

$$\lim_{t \to T^*} \frac{\|\boldsymbol{W}_i^{[1]}\|_2^2}{\boldsymbol{W}_{\boldsymbol{v},i}^2} = 1. \tag{29}$$

Combining Equations (28) and (29), we get

$$\lim_{t \to T^*} \cos\xi_{ij} = 1, \ i, j \in C_1. \tag{30}$$

This finishes the proof of statement (ii).

3. Finally, we calculate the norm $\|\dot{\boldsymbol{W}}^{[2]}\|_2^2$. By definition, we obtain

$$\langle\dot{\boldsymbol{W}}^{[2]}, \dot{\boldsymbol{W}}^{[2]}\rangle = \sum_{i=1}^{d_m}\sum_{j=1}^{d_m}\langle\boldsymbol{W}_{\boldsymbol{v},i}\boldsymbol{W}_i^{[1]}, \boldsymbol{W}_{\boldsymbol{v},j}\boldsymbol{W}_j^{[1]}\rangle.$$

We divide the sum into three parts due to the boundedness of entries of $\boldsymbol{W}_{\boldsymbol{v}}$.

$$\sum_{i\in C_1}\sum_{j\in C_1}\boldsymbol{W}_{\boldsymbol{v},i}^2\boldsymbol{W}_{\boldsymbol{v},j}^2\frac{\|\boldsymbol{W}_i^{[1]}\|_2}{|\boldsymbol{W}_{\boldsymbol{v},i}|}\frac{\|\boldsymbol{W}_j^{[1]}\|_2}{|\boldsymbol{W}_{\boldsymbol{v},j}|}\cos\xi_{ij} + 2\sum_{i\in C_1}\sum_{j\in C_2}\boldsymbol{W}_{\boldsymbol{v},i}\boldsymbol{W}_{\boldsymbol{v},j}\|\boldsymbol{W}_i^{[1]}\|_2\|\boldsymbol{W}_j^{[1]}\|_2\cos\xi_{ij}$$

$$+ \sum_{i\in C_2}\sum_{j\in C_2}\boldsymbol{W}_{\boldsymbol{v},i}\boldsymbol{W}_{\boldsymbol{v},j}\|\boldsymbol{W}_i^{[1]}\|_2\|\boldsymbol{W}_j^{[1]}\|_2\cos\xi_{ij}$$

$$= \sum_{i\in C_1}\sum_{j\in C_1}\boldsymbol{W}_{\boldsymbol{v},i}^2\boldsymbol{W}_{\boldsymbol{v},j}^2 + \sum_{i\in C_1}\sum_{j\in C_1}\boldsymbol{W}_{\boldsymbol{v},i}^2\boldsymbol{W}_{\boldsymbol{v},j}^2\left(\frac{\|\boldsymbol{W}_i^{[1]}\|_2}{|\boldsymbol{W}_{\boldsymbol{v},i}|}\frac{\|\boldsymbol{W}_j^{[1]}\|_2}{|\boldsymbol{W}_{\boldsymbol{v},j}|}\cos\xi_{ij} - 1\right)$$

$$+ 2\sum_{i\in C_1}\sum_{j\in C_2}\boldsymbol{W}_{\boldsymbol{v},i}\boldsymbol{W}_{\boldsymbol{v},j}\|\boldsymbol{W}_i^{[1]}\|_2\|\boldsymbol{W}_j^{[1]}\|_2\cos\xi_{ij} + \sum_{i\in C_2}\sum_{j\in C_2}\boldsymbol{W}_{\boldsymbol{v},i}\boldsymbol{W}_{\boldsymbol{v},j}\|\boldsymbol{W}_i^{[1]}\|_2\|\boldsymbol{W}_j^{[1]}\|_2\cos\xi_{ij}.$$

Since $\lim_{t\to T^*}\frac{\|\boldsymbol{W}_i^{[1]}\|_2}{|\boldsymbol{W}_{\boldsymbol{v},i}|}\frac{\|\boldsymbol{W}_j^{[1]}\|_2}{|\boldsymbol{W}_{\boldsymbol{v},j}|}\cos\xi_{ij} = 1$, we have

$$\lim_{t\to T^*}\frac{\langle\dot{\boldsymbol{W}}^{[2]}, \dot{\boldsymbol{W}}^{[2]}\rangle}{\left(\sum_{i\in C_1}\boldsymbol{W}_{\boldsymbol{v},i}^2\right)^2} = 1. \tag{31}$$

Based on statement (i), we obtain

$$\lim_{t\to T^*}\frac{\sum_{i\in C_1}\boldsymbol{W}_{\boldsymbol{v},i}^2}{\|\boldsymbol{W}_{\boldsymbol{v}}\|_2^2} = 1. \tag{32}$$

Combining Equations (31), (32) and conservation law, we have

$$\lim_{t \to T^*} \frac{\|\dot{\boldsymbol{W}}^{[2]}\|_2}{\|\boldsymbol{W}_{\boldsymbol{v}}\|_2^2} = \lim_{t \to T^*} \frac{\|\dot{\boldsymbol{W}}^{[2]}\|_2}{\|\boldsymbol{W}^{[2]}\|_2^2} = 1. \tag{33}$$

This finishes the proof of statement (iii). $\qquad\square$

Proposition 5 describes the angle between $\boldsymbol{W}_{\boldsymbol{v},i}\boldsymbol{W}_i^{[1]}$ and $\boldsymbol{W}_{\boldsymbol{v},j}\boldsymbol{W}_j^{[1]}$. Since $\dot{\boldsymbol{W}}^{[2]}$ is a linear combination of $\boldsymbol{W}_{\boldsymbol{v},i}\boldsymbol{W}_i^{[1]}$, we immediately have following corollary.

**Corollary 1.** *Suppose that Assumption 1 holds. Consider the effective dynamics Eq. (7) and recall index class defined in Proposition 5. The angle $\psi_i$ between the vectors $\dot{\boldsymbol{W}}^{[2]}$ and $\boldsymbol{W}_{\boldsymbol{v},i}\boldsymbol{W}_i^{[1]}$, as defined in Definition 5, satisfies:*

$$\lim_{t \to T^*} \cos \psi_i = 1, \quad i \in C_1. \tag{34}$$

*Proof.* By definition,

$$\cos \psi_i = \frac{\langle \boldsymbol{W}_{\boldsymbol{v},i}\boldsymbol{W}_i^{[1]}, \sum_{j=1}^m \boldsymbol{W}_{\boldsymbol{v},j}\boldsymbol{W}_j^{[1]} \rangle}{\|\boldsymbol{W}_{\boldsymbol{v},i}\boldsymbol{W}_i^{[1]}\|_2 \|\dot{\boldsymbol{W}}^{[2]}\|_2}.$$

Recall the definition of $\xi_{ij}$, the above equation can be reformulated as follows:

$$\cos \psi_i = \frac{\sum_{j=1}^m |\boldsymbol{W}_{\boldsymbol{v},i}||\boldsymbol{W}_{\boldsymbol{v},j}|\|\boldsymbol{W}_i^{[1]}\|_2\|\boldsymbol{W}_j^{[1]}\|_2 \cos \xi_{ij}}{\|\boldsymbol{W}_{\boldsymbol{v},i}\boldsymbol{W}_i^{[1]}\|_2 \|\dot{\boldsymbol{W}}^{[2]}\|_2}$$

$$= \frac{\sum_{j=1}^m |\boldsymbol{W}_{\boldsymbol{v},j}|\|\boldsymbol{W}_j^{[1]}\|_2 \cos \xi_{ij}}{\|\dot{\boldsymbol{W}}^{[2]}\|_2}$$

$$= \frac{\sum_{j \in C_1} \boldsymbol{W}_{\boldsymbol{v},j}^2 \frac{\|\boldsymbol{W}_j^{[1]}\|_2}{|\boldsymbol{W}_{\boldsymbol{v},j}|} \cos \xi_{ij} + \sum_{j \in C_2} |\boldsymbol{W}_{\boldsymbol{v},j}|\|\boldsymbol{W}_j^{[1]}\|_2 \cos \xi_{ij}}{\|\dot{\boldsymbol{W}}^{[2]}\|_2}.$$

According to Equations (21) and (22), we have $\lim_{t \to T^*} \cos \psi_i = 1$. This completes the proof. $\quad\square$

So far, we have characterized some properties of $\boldsymbol{W}_{\boldsymbol{v},i}\boldsymbol{W}_i^{[1]}$ which is component of $\dot{\boldsymbol{W}}^{[2]}$. We have shown that some of them will have the same direction when $t$ tends to $T^*$. However, it is not enough for our seek for a upper bound for energy $E$. Luckily, based on Corollary 1, we can analyze the angle between $\boldsymbol{W}^{[2]}$ and $\boldsymbol{W}_{\boldsymbol{v},i}\boldsymbol{W}_i^{[1]}$ which provides an upper bound. Before this, we give the following proposition. The subsequent proposition demonstrates an extension of statement 1 of Assumption 1, going beyond the condition of $\boldsymbol{W}_{\boldsymbol{v},i}\boldsymbol{W}_i^{[1]}\boldsymbol{W}^{[2]}$ being greater than zero to include additional angle-related information.

**Proposition 6.** *Suppose that Assumption 1 holds. Consider the effective dynamics Eq. (7) and recall index class defined in Proposition 5. There exists constants $T_1 \in (t_0, T^*)$ and $\Theta_1 \in [0, \frac{\pi}{2})$ such that for each index $i \in C_1$, the follow inequality holds:*

$$\cos \varphi_i \geq \cos \Theta_1, \quad t \in (T_1, T^*).$$

*Proof.* It is sufficient to prove the statement for any fixed $i \in C_1$ due to the finiteness of $|C_1|$. In Proposition 4, we have shown that $\langle \boldsymbol{W}_{\boldsymbol{v},i}\boldsymbol{W}_i^{[1]}, \boldsymbol{W}^{[2]} \rangle > 0$ for $t \in (t_0, T^*)$, which implies

$$\cos \varphi_i > 0, \quad t \in (t_0, T^*). \tag{35}$$

Hence we can focus on its square, i.e., $\cos^2 \varphi_i = \frac{\langle \boldsymbol{W}^{[2]}, \boldsymbol{W}_i^{[1]} \rangle^2}{\|\boldsymbol{W}^{[2]}\|_2^2\|\boldsymbol{W}_i^{[1]}\|_2^2}$. By direct calculation, the derivation of $\cos^2 \varphi_i$ is

$$\frac{2}{(\|\boldsymbol{W}^{[2]}\|_2^2\|\boldsymbol{W}_i^{[1]}\|_2^2)^2} \langle \boldsymbol{W}^{[2]}, \boldsymbol{W}_i^{[1]} \rangle (\langle \dot{\boldsymbol{W}}^{[2]}, \boldsymbol{W}_i^{[1]} \rangle + \langle \boldsymbol{W}^{[2]}, \dot{\boldsymbol{b}}^i \rangle)\|\boldsymbol{W}^{[2]}\|_2^2\|\boldsymbol{W}_i^{[1]}\|_2^2$$

$$- \frac{2}{(\|\boldsymbol{W}^{[2]}\|_2^2\|\boldsymbol{W}_i^{[1]}\|_2^2)^2} \langle \boldsymbol{W}^{[2]}, \boldsymbol{W}_i^{[1]} \rangle^2 (\langle \dot{\boldsymbol{W}}^{[2]}, \boldsymbol{W}^{[2]} \rangle\|\boldsymbol{W}_i^{[1]}\|_2^2 + \|\boldsymbol{W}^{[2]}\|_2^2\langle \dot{\boldsymbol{b}}^i, \boldsymbol{W}_i^{[1]} \rangle)$$

We can rewrite the numerator as

$$2\|\boldsymbol{W}_i^{[1]}\|_2^2\langle\boldsymbol{W}^{[2]},\boldsymbol{W}_i^{[1]}\rangle\left[\langle\dot{\boldsymbol{W}}^{[2]},\boldsymbol{W}_i^{[1]}\rangle\|\boldsymbol{W}^{[2]}\|_2^2-\langle\boldsymbol{W}^{[2]},\boldsymbol{W}_i^{[1]}\rangle\langle\dot{\boldsymbol{W}}^{[2]},\boldsymbol{W}^{[2]}\rangle\right]$$
$$+2\|\boldsymbol{W}^{[2]}\|_2^2\langle\boldsymbol{W}^{[2]},\boldsymbol{W}_{\boldsymbol{v},i}\boldsymbol{W}_i^{[1]}\rangle\left[\|\boldsymbol{W}^{[2]}\|_2^2\|\boldsymbol{W}_i^{[1]}\|_2^2-\langle\boldsymbol{W}^{[2]},\boldsymbol{W}_i^{[1]}\rangle^2\right].$$

The second term of above expression is obviously greater than zero by inequality of arithmetic and geometric means. Also we can rewrite the first term as

$$\frac{2\|\boldsymbol{W}_i^{[1]}\|_2^2}{\boldsymbol{W}_{\boldsymbol{v},i}^2}\langle\boldsymbol{W}^{[2]},\boldsymbol{W}_{\boldsymbol{v},i}\boldsymbol{W}_i^{[1]}\rangle\left[\langle\dot{\boldsymbol{W}}^{[2]},\boldsymbol{W}_{\boldsymbol{v},i}\boldsymbol{W}_i^{[1]}\rangle\|\boldsymbol{W}^{[2]}\|_2^2-\langle\boldsymbol{W}^{[2]},\boldsymbol{W}_{\boldsymbol{v},i}\boldsymbol{W}_i^{[1]}\rangle\langle\dot{\boldsymbol{W}}^{[2]},\boldsymbol{W}^{[2]}\rangle\right].$$

We find that the first two factors $\frac{2\|\boldsymbol{W}_i^{[1]}\|_2^2}{\boldsymbol{W}_{\boldsymbol{v},i}^2}\langle\boldsymbol{W}^{[2]},\boldsymbol{W}_{\boldsymbol{v},i}\boldsymbol{W}_i^{[1]}\rangle$ of above expression are positive. According to Definition 5, the difference term can be reformulated as

$$\langle\dot{\boldsymbol{W}}^{[2]},\boldsymbol{W}_{\boldsymbol{v},i}\boldsymbol{W}_i^{[1]}\rangle\|\boldsymbol{W}^{[2]}\|^2-\langle\boldsymbol{W}^{[2]},\boldsymbol{W}_{\boldsymbol{v},i}\boldsymbol{W}_i^{[1]}\rangle\langle\dot{\boldsymbol{W}}^{[2]},\boldsymbol{W}^{[2]}\rangle$$
$$=\|\boldsymbol{W}^{[2]}\|_2^2\|\dot{\boldsymbol{W}}^{[2]}\|_2\|\boldsymbol{W}_{\boldsymbol{v},i}\boldsymbol{W}_i^{[1]}\|_2(\cos\psi_i-\cos\zeta\cos\varphi_i).$$

Note that $\lim_{t\to T^*}\cos\psi_i=1$ for $i\in C_1$. So for every $\varepsilon>0$, there exists $\delta>0$ such that

$$1-\varepsilon\le\cos\psi_i\le 1,\quad t\in(T^*-\delta,T^*).$$

Set $\bar{t}_i=T^*-\delta$ and $\bar{\theta}_i=\arccos(1-\varepsilon)$. Then we have either $\cos\varphi_i\ge\cos\bar{\theta}_i$, $t\in(\bar{t}_i,T^*)$, or there exists $t\in(\bar{t}_i,T^*)$ such that $\cos\varphi_i\le\cos\bar{\theta}_i$, then it will increase monotonically until it goes up to $\bar{\theta}_i$. No matter in which case, we can find $\tilde{\theta}_i\in[0,\frac{\pi}{2})$ such that $\cos\varphi_i\ge\cos\tilde{\theta}_i$. Let $T_1=\max_{i\in C_1}\bar{t}_i$ and $\Theta_1=\max_{i\in C_1}\tilde{\theta}_i$. Thus, for each $i\in C_1$, the following inequality holds

$$\cos\varphi_i\ge\cos\Theta_1,\quad t\in(T_1,T^*).$$

This completes the proof. $\qquad\square$

In fact, Proposition 6 provides the angular relationship between $\boldsymbol{W}^{[2]}$ and its derivative $\dot{\boldsymbol{W}}^{[2]}$. We summarize it as follows.

**Proposition 7.** *Suppose that Assumption 1 holds. Consider the effective dynamics Eq. (7). There exists $T_2\in(T_1,T^*)$ and $\Theta_2\in[0,\frac{\pi}{2})$ such that*

$$\cos\zeta\ge\cos\Theta_2,\quad t\in(T_2,T^*). \tag{36}$$

*Moreover, recall the definition of energy $E$, the following inequality holds*

$$E=\langle\boldsymbol{W}^{[2]},\dot{\boldsymbol{W}}^{[2]}\rangle\ge\cos\Theta_2\|\boldsymbol{W}^{[2]}\|_2\|\dot{\boldsymbol{W}}^{[2]}\|_2,\quad t\in(T_2,T^*). \tag{37}$$

*Proof.* By definition of $\zeta$, we obtain

$$\cos\zeta=\frac{\langle\dot{\boldsymbol{W}}^{[2]},\boldsymbol{W}^{[2]}\rangle}{\|\boldsymbol{W}^{[2]}\|_2\|\dot{\boldsymbol{W}}^{[2]}\|_2}=\frac{\langle\sum_{j=1}^m\boldsymbol{W}_{\boldsymbol{v},i}\boldsymbol{W}_i^{[1]},\boldsymbol{W}^{[2]}\rangle}{\|\boldsymbol{W}^{[2]}\|_2\|\dot{\boldsymbol{W}}^{[2]}\|_2}.$$

According to Proposition 6, we have

$$\cos\zeta\ge\cos\Theta_1\frac{\sum_{i\in C_1}\|\boldsymbol{W}_{\boldsymbol{v},i}\boldsymbol{W}_i^{[1]}\|_2}{\|\dot{\boldsymbol{W}}^{[2]}\|_2},\quad t\in(T_1,T^*).$$

Since we have shown that $\lim_{t\to T^*}\frac{\sum_{i\in C_1}\|\boldsymbol{W}_{\boldsymbol{v},i}\boldsymbol{W}_i^{[1]}\|_2}{\|\dot{\boldsymbol{W}}^{[2]}\|_2}=1$ according to the proof of Proposition 5, there exists $T_2\in(T_1,T^*)$ and $\Theta_2\in[0,\frac{\pi}{2})$ such that

$$\cos\zeta\ge\cos\Theta_2,\quad t\in(T_2,T^*).$$

Recall the definition of energy $E$, we have

$$E=\langle\boldsymbol{W}^{[2]},\dot{\boldsymbol{W}}^{[2]}\rangle\ge(\cos\Theta_2)\|\boldsymbol{W}^{[2]}\|_2\|\dot{\boldsymbol{W}}^{[2]}\|_2,\quad t\in(T_2,T^*).$$

This completes the proof. $\qquad\square$

Now we prove the proposition with all preparations above.

**Proposition 8** (energy upper bound and blow up estimate). *Suppose that Assumption 1 holds. Consider the effective dynamics Eq. (7). There exist $T_3 \in (T_2, T^*)$ and $C \geq 1$ such that the following upper bound of Energy $E$ holds*

$$E(t) \leq \frac{1}{\left(E(s)^{-\frac{1}{3}} - C(t-s)\right)^3}, \quad T_3 \leq s < t < T^*. \tag{38}$$

*Moreover, the blow up time $T^*$ is bounded below by*

$$T^* \geq t + \frac{E(t)^{-\frac{1}{3}}}{C}, \quad t < T^*. \tag{39}$$

*Proof.* First, by calculating the derivative of energy $E$, we have

$$\dot{E} = \|\dot{\boldsymbol{W}}^{[2]}\|_2^2 + \|\dot{\boldsymbol{W}}_{\boldsymbol{v}}\|_2^2 + \|\boldsymbol{W}^{[2]}\|_2^2 \|\boldsymbol{W}_{\boldsymbol{v}}\|_2^2.$$

We rewrite the derivative of energy $E$ as

$$\dot{E} = \|\dot{\boldsymbol{W}}^{[2]}\|_2^2 \left(1 + \frac{\|\dot{\boldsymbol{W}}_{\boldsymbol{v}}\|_2^2}{\|\dot{\boldsymbol{W}}^{[2]}\|_2^2} + \frac{\|\boldsymbol{W}^{[2]}\|_2^2 \|\boldsymbol{W}_{\boldsymbol{v}}\|_2^2}{\|\dot{\boldsymbol{W}}^{[2]}\|_2^2}\right)$$

$$= \|\dot{\boldsymbol{W}}^{[2]}\|_2^{\frac{4}{3}} \|\boldsymbol{W}^{[2]}\|_2^{\frac{4}{3}} \frac{\|\dot{\boldsymbol{W}}^{[2]}\|_2^{\frac{2}{3}}}{\|\boldsymbol{W}^{[2]}\|_2^{\frac{4}{3}}} \left(1 + \frac{\|\dot{\boldsymbol{W}}_{\boldsymbol{v}}\|_2^2}{\|\dot{\boldsymbol{W}}^{[2]}\|_2^2} + \frac{\|\boldsymbol{W}^{[2]}\|_2^2 \|\boldsymbol{W}_{\boldsymbol{v}}\|_2^2}{\|\dot{\boldsymbol{W}}^{[2]}\|_2^2}\right).$$

According to conservation laws and Equation (22), there exists $T_3 > T_2$ such that

$$\frac{\|\dot{\boldsymbol{W}}^{[2]}\|_2^{\frac{2}{3}}}{\|\boldsymbol{W}^{[2]}\|_2^{\frac{4}{3}}} \left(1 + \frac{\|\dot{\boldsymbol{W}}_{\boldsymbol{v}}\|_2^2}{\|\dot{\boldsymbol{W}}^{[2]}\|_2^2} + \frac{\|\boldsymbol{W}^{[2]}\|_2^2 \|\boldsymbol{W}_{\boldsymbol{v}}\|_2^2}{\|\dot{\boldsymbol{W}}^{[2]}\|_2^2}\right) \leq 4.$$

Then we have

$$\dot{E} \leq 4 \|\dot{\boldsymbol{W}}^{[2]}\|_2^{\frac{4}{3}} \|\boldsymbol{W}^{[2]}\|_2^{\frac{4}{3}}, \quad \forall t > T_3.$$

Based on Proposition 7, we have

$$\|\boldsymbol{W}^{[2]}\|_2 \|\dot{\boldsymbol{W}}^{[2]}\|_2 \leq \frac{1}{\cos\Theta_2} E, \quad \forall t > T_3.$$

Thus we obtain

$$\dot{E} \leq 4(\frac{1}{\cos\Theta_2})^{\frac{4}{3}} E, \quad \forall t > T_3.$$

We denote $4(\frac{1}{\cos\Theta_2})^{\frac{4}{3}}$ as $C_0$. Then we have $C_0 \geq 4$ and

$$\dot{E} \leq C_0 E^{\frac{4}{3}}.$$

Opposite to proof of Theorem 1, we obtain

$$\frac{\mathrm{d}}{\mathrm{d}t} E^{-\frac{1}{3}} \geq -\frac{1}{3} C_0, \quad \forall t > T_3.$$

We denote $\frac{C_0}{3}$ as $C$. Thus $C \geq 1$ and we have

$$E(t) \leq \frac{1}{\left(E(s)^{-\frac{1}{3}} - C(t-s)\right)^3}, \quad T_3 < s < t < T^*. \tag{40}$$

Hence, for each time $t < T^*$, the time of blow up is bounded below by

$$T^* \geq t + \frac{E(t)^{-\frac{1}{3}}}{C}. \tag{41}$$

This completes the proof. $\qquad\square$

Now we begin the proof for Theorem 2.

*Proof.* We just prove the case for $\boldsymbol{W}_{\boldsymbol{v},i} > 0$. And the case for $\boldsymbol{W}_{\boldsymbol{v},i} < 0$ follows by similar argument. Because the derivative of $\boldsymbol{W}_{\boldsymbol{v},i}$ is $\boldsymbol{W}_i^{[1]}\boldsymbol{W}^{[2]}$, we only need to show that

$$\int_{T_3}^{T^*} \boldsymbol{W}_i^{[1]}\boldsymbol{W}^{[2]}\,\mathrm{d}t = +\infty. \tag{42}$$

Since $\lim_{t\to T^*}\frac{\|\boldsymbol{W}^{[2]}\|_2^2}{\|\dot{\boldsymbol{W}}^{[2]}\|_2} = 1$ due to the statement (iii) of Proposition 5, there exists $T_4 > T_3$ such that $\|\dot{\boldsymbol{W}}^{[2]}\|_2 \le 2\sqrt{2}\|\boldsymbol{W}^{[2]}\|_2^2$, which implies

$$E = \langle \dot{\boldsymbol{W}}^{[2]}, \boldsymbol{W}^{[2]}\rangle \le \|\dot{\boldsymbol{W}}^{[2]}\|_2\|\boldsymbol{W}^{[2]}\|_2 \le 2\sqrt{2}\|\boldsymbol{W}^{[2]}\|_2^3. \tag{43}$$

The idea is we can find a infinite division of $(T_4, T^*)$ such that the integral of $\boldsymbol{W}_i^{[1]}\boldsymbol{W}^{[2]}$ on each sub-interval is larger than positive constant which is an independent constant. Then we consider the integral $\int_{t_1}^{t_2} \boldsymbol{W}_i^{[1]}\boldsymbol{W}^{[2]}\mathrm{d}t$. By direct calculation of the derivative of $\boldsymbol{W}_i^{[1]}\boldsymbol{W}^{[2]}$, we have

$$(\boldsymbol{W}_i^{[1]}\boldsymbol{W}^{[2]})\dot{} = \boldsymbol{W}_{\boldsymbol{v},i}\boldsymbol{W}^{[2]\mathsf{T}}\boldsymbol{W}^{[2]} + \boldsymbol{W}_i^{[1]}\left(\sum_j \boldsymbol{W}_{\boldsymbol{v},j}\boldsymbol{W}_j^{[1]}\right).$$

Integrating both sides of the equality, we have

$$(\boldsymbol{W}_i^{[1]}\boldsymbol{W}^{[2]})(t) = (\boldsymbol{W}_i^{[1]}\boldsymbol{W}^{[2]})(t_1) + \int_{t_1}^{t}\boldsymbol{W}_{\boldsymbol{v},i}\boldsymbol{W}^{[2]\mathsf{T}}\boldsymbol{W}^{[2]} + \boldsymbol{W}_i^{[1]}\left(\sum_j \boldsymbol{W}_{\boldsymbol{v},j}\boldsymbol{W}_j^{[1]}\right)\mathrm{d}s$$

$$\ge \boldsymbol{W}_{\boldsymbol{v},i}(T_4)\int_{t_1}^{t}\boldsymbol{W}^{[2]\mathsf{T}}\boldsymbol{W}^{[2]}\mathrm{d}s.$$

Note that Eq. (43) implies

$$\int_{t_1}^{t}\boldsymbol{W}^{[2]\mathsf{T}}\boldsymbol{W}^{[2]}\mathrm{d}s \ge \frac{1}{2}\int_{t_1}^{t}E^{\frac{2}{3}}(s)\mathrm{d}s$$

$$\ge \frac{1}{2}\int_{t_1}^{t}\frac{1}{(E(t_1)^{-\frac{1}{3}} - (s - t_1))^2}\mathrm{d}s$$

$$= \frac{1}{2}\left[\frac{1}{E(t_1)^{-\frac{1}{3}} - (t - t_1)} - \frac{1}{E(t_1)^{-\frac{1}{3}}}\right].$$

Thus the integral satisfies

$$\int_{t_1}^{t_2}\boldsymbol{W}_i^{[1]}\boldsymbol{W}^{[2]}\mathrm{d}t \ge \boldsymbol{W}_{\boldsymbol{v},i}(T_4)\frac{1}{2}\int_{t_1}^{t_2}\frac{1}{E(t_1)^{-\frac{1}{3}} - (t - t_1)} - \frac{1}{E(t_1)^{-\frac{1}{3}}}\mathrm{d}t$$

$$= \boldsymbol{W}_{\boldsymbol{v},i}(T_4)\frac{1}{2}\left[-\ln(E(t_1)^{-\frac{1}{3}} - (t_2 - t_1)) + \ln(E(t_1)^{-\frac{1}{3}}) - \frac{t_2 - t_1}{E(t_1)^{-\frac{1}{3}}}\right].$$

According to Theorem 8, we can choose $t_2 - t_1 = \frac{E(t_1)^{-\frac{1}{3}}}{2C}$. Thus we obtain

$$\int_{t_1}^{t_2}\boldsymbol{W}_i^{[1]}\boldsymbol{W}^{[2]}\mathrm{d}t \ge \boldsymbol{W}_{\boldsymbol{v},i}(T_4)\frac{1}{2}\left[\ln\frac{1}{1 - \frac{1}{2C}} - \frac{1}{2C}\right]. \tag{44}$$

Then we introduce auxiliary function

$$f(t) = \ln\frac{1}{1 - t} - t$$

$$= -\ln(1 - t) - t.$$

We have $f(0) = 0$ and $\dot{f}(t) > 0$. Then we have $\ln\frac{1}{1 - \frac{1}{2C}} - \frac{1}{2C} > 0$. Since there are infinitely many such sub-intervals, the proof of the theorem is completed. $\qquad\square$

### A.3 Theory details for Key-Query Dynamics

This appendix provides the detailed derivations for Proposition 3 and Theorem 3, assuming a linear activation function and holding to Assumption 2 (or its empirical variant, Assumption 2*). Our approach begins with a standard asymptotic analysis to decompose the loss function, as shown in Eq. (13).

Starting from the definition of the empirical risk, we have:

$$
\begin{aligned}
\mathcal{L}(\boldsymbol{\theta}) &= \frac{1}{n} \sum_{i=1}^{n} e^{-y_i f_{\boldsymbol{\theta}}(\boldsymbol{X}_i)_s} \\
&= \frac{1}{n} \sum_{i=1}^{n} \exp\left(-y_i \left(\sum_{j=1}^{s} \frac{\exp\left(\frac{\boldsymbol{X}_{i,s} \boldsymbol{W}_Q \boldsymbol{W}_K^\intercal \boldsymbol{X}_{i,j}^\intercal}{\sqrt{d_m}}\right)}{\sum_{l=1}^{s} \exp\left(\frac{\boldsymbol{X}_{i,s} \boldsymbol{W}_Q \boldsymbol{W}_K^\intercal \boldsymbol{X}_{i,l}^\intercal}{\sqrt{d_m}}\right)} \boldsymbol{X}_{i,j} \boldsymbol{W}_V \boldsymbol{W}^{[1]} \boldsymbol{W}^{[2]}\right)\right) \\
&= \frac{1}{n} \sum_{i=1}^{n} \exp\left(-y_i \sum_{j=1}^{s} \left[\left(\frac{1}{s} + \frac{1}{s} \frac{\boldsymbol{X}_{i,s} \boldsymbol{W}_Q \boldsymbol{W}_K^\intercal \boldsymbol{X}_{i,j}^\intercal}{\sqrt{d_m}}\right.\right.\right. \\
&\qquad\qquad\qquad \left.\left.\left. - \frac{1}{s^2} \sum_{l=1}^{s} \frac{\boldsymbol{X}_{i,s} \boldsymbol{W}_Q \boldsymbol{W}_K^\intercal \boldsymbol{X}_{i,l}^\intercal}{\sqrt{d_m}} + \mathcal{O}(\delta^4)\right) \boldsymbol{X}_{i,j} \boldsymbol{W}_V \boldsymbol{W}^{[1]} \boldsymbol{W}^{[2]}\right]\right).
\end{aligned}
\tag{45}
$$

The third equality results from applying a Taylor expansion to the softmax function, which is justified by Assumption 2. The higher-order term, $\mathcal{O}(\delta^4)$, is subsequently omitted as it does not affect the leading-order training dynamics. Given the property of the exponential loss function, $\ell(q) = e^{-q}$, the empirical loss can be decomposed as follows:

$$
\begin{aligned}
\mathcal{L}(\boldsymbol{\theta}) &= \frac{1}{n} \sum_{i=1}^{n} \exp\left(-y_i \left(\sum_{j=1}^{s} \frac{1}{s} \boldsymbol{X}_{i,j} \boldsymbol{W}_V \boldsymbol{W}^{[1]} \boldsymbol{W}^{[2]}\right)\right) \\
&\quad \cdot \exp\left(-y_i \left(\sum_{j=1}^{s} \left(\frac{1}{s} \frac{\boldsymbol{X}_{i,s} \boldsymbol{W}_Q \boldsymbol{W}_K^\intercal \boldsymbol{X}_{i,j}^\intercal}{\sqrt{d_m}} - \frac{1}{s^2} \sum_{l=1}^{s} \frac{\boldsymbol{X}_{i,s} \boldsymbol{W}_Q \boldsymbol{W}_K^\intercal \boldsymbol{X}_{i,l}^\intercal}{\sqrt{d_m}}\right) \boldsymbol{X}_{i,j} \boldsymbol{W}_V \boldsymbol{W}^{[1]} \boldsymbol{W}^{[2]}\right)\right) \\
&= \frac{1}{n} \sum_{i=1}^{n} \mathcal{L}_{1,i}(\boldsymbol{\theta}) \mathcal{L}_{2,i}(\boldsymbol{\theta}),
\end{aligned}
\tag{46}
$$

where $\mathcal{L}_{1,i}(\boldsymbol{\theta})$ and $\mathcal{L}_{2,i}(\boldsymbol{\theta})$ correspond to the two exponential factors in the preceding expression.

#### A.3.1 Proof for Proposition 3 under Assumption 2

*Proof.* The proof is structured in two parts. First, we demonstrate the separation of dynamics by showing that the gradients with respect to different sets of weights have different orders of magnitude. Second, we derive the specific dynamics for the key and query matrices.

**Dynamics Separation**: By symmetry, we will detail the calculations for the partial derivatives with respect to $\boldsymbol{W}_V$ and $\boldsymbol{W}_Q$. The derivations for the other weight matrices follow a similar procedure.

We begin by computing the partial derivative of the loss $\mathcal{L}$ with respect to $\boldsymbol{W}_V$. Applying the product rule to the decomposed loss from Eq. (46), we obtain:

$$
\begin{aligned}
\frac{\partial \mathcal{L}}{\partial \boldsymbol{W}_V} &= \frac{\partial}{\partial \boldsymbol{W}_V} \left(\frac{1}{n} \sum_{i=1}^{n} \mathcal{L}_{1,i}(\boldsymbol{\theta}) \mathcal{L}_{2,i}(\boldsymbol{\theta})\right) \\
&= \frac{1}{n} \sum_{i=1}^{n} \left(\frac{\partial \mathcal{L}_{1,i}}{\partial \boldsymbol{W}_V} \mathcal{L}_{2,i} + \mathcal{L}_{1,i} \frac{\partial \mathcal{L}_{2,i}}{\partial \boldsymbol{W}_V}\right) \\
&= \frac{1}{n} \sum_{i=1}^{n} \left(\frac{\partial \mathcal{L}_{1,i}}{\partial \boldsymbol{W}_V} \left(1 + \mathcal{O}(\delta^2)\right) + \mathcal{L}_{1,i} \cdot \mathcal{O}(\delta^2)\right).
\end{aligned}
\tag{47}
$$

The third equality holds based on Assumption 2, which implies that $\mathcal{L}_{2,i} = 1 + \mathcal{O}(\delta^2)$ and its derivative $\frac{\partial \mathcal{L}_{2,i}}{\partial W_V}$ is also of order $\mathcal{O}(\delta^2)$. Furthermore, Assumption 2 states that the leading-order term of the loss, $\mathcal{L}_{1,i}$, is independent of the attention mechanism weights at initialization. Consequently, the term $\frac{\partial \mathcal{L}_{1,i}}{\partial W_V}$ is of order $\mathcal{O}(\delta^2)$. This implies that the entire gradient $\frac{\partial \mathcal{L}}{\partial W_V}$ is dominated by terms of order $\mathcal{O}(\delta^2)$.

Next, we consider the partial derivative with respect to $W_Q$:

$$
\begin{aligned}
\frac{\partial \mathcal{L}}{\partial W_Q} &= \frac{\partial}{\partial W_Q}\left(\frac{1}{n}\sum_{i=1}^{n}\mathcal{L}_{1,i}(\boldsymbol{\theta})\mathcal{L}_{2,i}(\boldsymbol{\theta})\right) \\
&= \frac{1}{n}\sum_{i=1}^{n}\mathcal{L}_{1,i}(\boldsymbol{\theta})\frac{\partial \mathcal{L}_{2,i}}{\partial W_Q},
\end{aligned}
\tag{48}
$$

since $\mathcal{L}_{1,i}$ is independent of $W_Q$. Based on Assumption 2, the term $\frac{\partial \mathcal{L}_{2,i}}{\partial W_Q}$ is of order $\mathcal{O}(\delta)$, which establishes that the overall gradient $\frac{\partial \mathcal{L}}{\partial W_Q}$ is also of order $\mathcal{O}(\delta)$.

**Key-Query Dynamics**: The principle of dynamics separation, established above, shows that the gradients with respect to $W_Q$ and $W_K$ (order $\mathcal{O}(\delta)$) are significantly larger than those for $W_V$, $W^{[1]}$, and $W^{[2]}$ (order $\mathcal{O}(\delta^2)$). Therefore, during the initial phase of training, the dynamics are dominated by the updates to $W_Q$ and $W_K$. We can thus analyze their leading-order dynamics by treating the other weight matrices as effectively constant.

To derive these dynamics, we employ matrix calculus with differentials. For a scalar function $f(\mathbf{X})$ of a matrix variable $\mathbf{X}$, the differential is given by $\mathrm{d}f = \mathrm{tr}\left(\left(\frac{\partial f}{\partial \mathbf{X}}\right)^{\mathsf{T}}\mathrm{d}\mathbf{X}\right)$. We apply this to the argument of the exponential in $\mathcal{L}_{2,i}$, which we denote as $A_i(\boldsymbol{\theta})$. The differential of $A_i$ with respect to $W_Q$ is:

$$
\begin{aligned}
&\mathrm{d}\left(-y_i\left(\sum_{j=1}^{s}\left(\frac{1}{s}\frac{X_{i,s}W_Q W_K^{\mathsf{T}}X_{i,j}^{\mathsf{T}}}{\sqrt{d_m}} - \frac{1}{s^2}\sum_{l=1}^{s}\frac{X_{i,s}W_Q W_K^{\mathsf{T}}X_{i,l}^{\mathsf{T}}}{\sqrt{d_m}}\right)X_{i,j}W_V W^{[1]}W^{[2]}\right)\right) \\
&= -\frac{y_i}{s\sqrt{d_m}}X_{i,s}(\mathrm{d}W_Q)W_K^{\mathsf{T}}\left[\sum_{j=1}^{s}\left(X_{i,j}^{\mathsf{T}} - \frac{1}{s}\sum_{l=1}^{s}X_{i,l}^{\mathsf{T}}\right)X_{i,j}\right]W_V W^{[1]}W^{[2]} \\
&= \mathrm{tr}\left(-\frac{y_i}{s\sqrt{d_m}}W_K\left[\sum_{j=1}^{s}X_{i,j}^{\mathsf{T}}\left(X_{i,j} - \frac{1}{s}\sum_{l=1}^{s}X_{i,l}\right)\right]W_V W^{[1]}W^{[2]}X_{i,s}^{\mathsf{T}}(\mathrm{d}W_Q)^{\mathsf{T}}\right).
\end{aligned}
\tag{49}
$$

By identifying the coefficient of $\mathrm{d}W_Q$ from the trace form, we obtain the gradient. Consequently, after neglecting higher-order terms, the leading-order dynamics for $W_Q$ under the gradient flow $\frac{\mathrm{d}W_Q}{\mathrm{d}t} = -\frac{\partial \mathcal{L}}{\partial W_Q}$ are given by:

$$
\frac{\mathrm{d}W_Q}{\mathrm{d}t} = \frac{1}{ns\sqrt{d_m}}\sum_{i=1}^{n}y_i\mathcal{L}_{1,i}X_{i,s}^{\mathsf{T}}\left(W_V W^{[1]}W^{[2]}\right)^{\mathsf{T}}\left[\sum_{j=1}^{s}X_{i,j}^{\mathsf{T}}\left(X_{i,j} - \frac{1}{s}\sum_{l=1}^{s}X_{i,l}\right)\right]^{\mathsf{T}}W_K.
\tag{50}
$$

A symmetric argument yields the corresponding dynamics for $W_K$. This completes the proof. $\square$

## A.4 Proof for Proposition 3 under Assumption 2*

*Proof.* **Dynamics Separation**: We use another scheme to estimate the gradient. Consider

$$
\begin{aligned}
\frac{\partial \mathcal{L}}{\partial \boldsymbol{W}_V} &= \frac{\partial}{\partial \boldsymbol{W}_V} \left( \frac{1}{n} \sum_{i=1}^{n} \mathcal{L}_{1,i}(\boldsymbol{\theta}) \mathcal{L}_{2,i}(\boldsymbol{\theta}) \right) \\
&= \frac{1}{n} \sum_{i=1}^{n} \left( \frac{\partial \mathcal{L}_{1,i}}{\partial \boldsymbol{W}_V} \mathcal{L}_{2,i} + \mathcal{L}_{1,i} \frac{\partial \mathcal{L}_{2,i}}{\partial \boldsymbol{W}_V} \right) \\
&= \frac{1}{n} \sum_{i=1}^{n} \left( \frac{\partial \mathcal{L}_{1,i}}{\partial \boldsymbol{W}_V} \left( 1 + \mathcal{O}(\delta^2) \right) + \mathcal{L}_{1,i} \cdot \mathcal{O}(\delta^2) \right) \\
&= \frac{1}{n} \sum_{i=1}^{n} \frac{\partial \mathcal{L}_{1,i}}{\partial \boldsymbol{W}_V} + \mathcal{O}(\delta^2).
\end{aligned}
\tag{51}
$$

Take the inner product of both sides of the equation with $\frac{\partial \mathcal{L}}{\partial \boldsymbol{W}_V}$, we have

$$
\left\langle \frac{\partial \mathcal{L}}{\partial \boldsymbol{W}_V}, \frac{\partial \mathcal{L}}{\partial \boldsymbol{W}_V} \right\rangle = \left\langle \frac{1}{n} \sum_{i=1}^{n} \frac{\partial \mathcal{L}_{1,i}}{\partial \boldsymbol{W}_V}, \frac{\partial \mathcal{L}}{\partial \boldsymbol{W}_V} \right\rangle + \mathcal{O}(\delta^2).
\tag{52}
$$

However, note that

$$
\frac{\mathrm{d}}{\mathrm{d}t} \widetilde{\mathcal{L}}_1 = - \left\langle \frac{1}{n} \sum_{i=1}^{n} \frac{\partial \mathcal{L}_{1,i}}{\partial \boldsymbol{W}_V}, \frac{\partial \mathcal{L}}{\partial \boldsymbol{W}_V} \right\rangle - \left\langle \frac{1}{n} \sum_{i=1}^{n} \frac{\partial \mathcal{L}_{1,i}}{\partial \boldsymbol{W}^{[1]}}, \frac{\partial \mathcal{L}}{\partial \boldsymbol{W}^{[1]}} \right\rangle - \left\langle \frac{1}{n} \sum_{i=1}^{n} \frac{\partial \mathcal{L}_{1,i}}{\partial \boldsymbol{W}^{[2]}}, \frac{\partial \mathcal{L}}{\partial \boldsymbol{W}^{[2]}} \right\rangle.
\tag{53}
$$

Based on Assumption 2*, we have $\frac{\partial \mathcal{L}}{\partial \boldsymbol{W}_V}$ is $\mathcal{O}(\delta^2)$. The rest of the proof is similar to the previous one.

**Key-Query Dynamics**: This part is almost the same. Just need to note that the condition $\frac{\mathrm{d}}{\mathrm{d}t} \left( \frac{\boldsymbol{W}_V}{\|\boldsymbol{W}_V\|} \right) = \frac{\mathrm{d}}{\mathrm{d}t} \left( \frac{\boldsymbol{W}^{[1]}}{\|\boldsymbol{W}^{[1]}\|} \right) = \frac{\mathrm{d}}{\mathrm{d}t} \left( \frac{\boldsymbol{W}^{[2]}}{\|\boldsymbol{W}^{[2]}\|} \right) \approx \boldsymbol{0}$ gives the same $\boldsymbol{F}$ after normalization.

$\square$

## A.5 Proof for Theorem 3

*Proof.* Based on the leading-order dynamics of key-query matrices, we prove the theorem for $\boldsymbol{W}_Q$ and the technique for $\boldsymbol{W}_K$ is similar. Differentiating the dynamics again, we obtain:

$$
\frac{\mathrm{d}^2}{\mathrm{d}t^2} \boldsymbol{W}_Q = \boldsymbol{F}\boldsymbol{F}^\mathsf{T} \boldsymbol{W}_Q.
\tag{54}
$$

Let the singular value decomposition of $\boldsymbol{F}$ be $\boldsymbol{F} = \boldsymbol{U}\boldsymbol{\Sigma}\boldsymbol{V}^\mathsf{T}$, then the dynamics can be rewritten as

$$
\frac{\mathrm{d}^2}{\mathrm{d}t^2} \boldsymbol{W}_Q = \boldsymbol{U}\boldsymbol{\Sigma}^2\boldsymbol{U}^\mathsf{T} \boldsymbol{W}_Q.
\tag{55}
$$

Let $\tilde{\boldsymbol{W}}_Q = \boldsymbol{U}^\mathsf{T} \boldsymbol{W}_Q \boldsymbol{U}$, we have

$$
\frac{\mathrm{d}^2}{\mathrm{d}t^2} \tilde{\boldsymbol{W}}_Q = \boldsymbol{\Sigma}^2 \tilde{\boldsymbol{W}}_Q.
\tag{56}
$$

The evolutions of entries are

$$
\tilde{\boldsymbol{W}}_{Q,ij}(t) = C_{1,ij} \, \mathrm{e}^{\lambda_i t} + C_{2,ij} \, \mathrm{e}^{-\lambda_i t}.
\tag{57}
$$

As a result,

$$
\mathrm{rank} \left( \lim_{t \to \infty} \frac{\boldsymbol{W}_Q}{\|\boldsymbol{W}_Q\|_\mathrm{F}} \right) \le k,
\tag{58}
$$

where $k$ is the multiplicity of the largest singular value. This result finishes the proof.

# B  Experimental Details

In this section, we present more experimental details to supplement the main text.

## B.1  Experimental Setting of Synthetic Dataset

We introduce the dataset construction method and training hyperparameters used to train the synthetic dataset.

First, we give some calculation methods for experimental pictures.

**Satisfaction rate.** We denote the conditions in Assumption 1 as follows:

$$A_1 = \left\{ i \in [d_m] \ \Big| \ \boldsymbol{W}_i^{[2]} \boldsymbol{W_v} \boldsymbol{W}^{[1],i} > 0, \boldsymbol{W_{v,i}} \boldsymbol{W}_i^{[1]} \boldsymbol{W}^{[2]} > 0 \right\},$$

$$A_2 = \left\{ (i,j) \in [d_m] \times [d_m] \ \Big| \ \langle \boldsymbol{W}_i^{[2]} \boldsymbol{W}^{[1],i}, \ \boldsymbol{W}_j^{[2]} \boldsymbol{W}^{[1],j} \rangle > 0, \langle \boldsymbol{W_{v,i}} \boldsymbol{W}^{[1],i}, \boldsymbol{W_{v,j}} \boldsymbol{W}^{[1],j} \rangle > 0 \right\}.$$

**Cosine similarity.** To visualize the internal structure of a weight matrix $\mathbf{W}$, we generate a heatmap of its reordered row-wise cosine similarity matrix. The procedure is as follows: first, the row-wise cosine similarity matrix $\mathbf{S}$ is computed, where each entry $S_{ij} = \cos(\mathbf{w}_i, \mathbf{w}_j)$ measures the similarity between row vectors $\mathbf{w}_i$ and $\mathbf{w}_j$.

To reveal underlying block structures, we then employ a spectral reordering technique. This involves finding the principal eigenvector $\mathbf{v}_{\max}$ (the one corresponding to the largest eigenvalue) of the similarity matrix $\mathbf{S}$. The sorted order of this eigenvector's components, $\mathcal{P} = \text{argsort}(\mathbf{v}_{\max})$, provides a permutation index. By applying this permutation to both the rows and columns of $\mathbf{S}$, we group highly correlated row vectors together, making low-rank patterns visually apparent in the final heatmap.

**Dataset Construction.** We construct synthetic datasets using the concept of the anchor function Zhang et al. [2024c], which enables controlled simulation of linguistic relationships. Let the set of prompt anchors be $\mathcal{A} = \{a \in \mathbb{N}^+ \mid \alpha_{\min} \leq a \leq \alpha_{\max}\}$ and the set of keys be $\mathcal{Z} = \{z \in \mathbb{N}^+ \mid \zeta_{\min} \leq z \leq \zeta_{\max}\}$, where $\mathcal{A}$ and $\mathcal{Z}$ are disjoint, i.e., $\mathcal{A} \cap \mathcal{Z} = \emptyset$.

We define an anchor function $\mathcal{F}(\boldsymbol{X}) : \mathbb{N}^s \to \mathbb{N}$, where $\boldsymbol{X} = (x_1, x_2, \ldots, x_s)$ is a sequence of length $s$. Each sequence contains exactly one anchor token $a \in \mathcal{A}$ among the first $s - 1$ positions, and the function outputs the token immediately following the anchor, shifted by $a$:

$$\mathcal{F}(x_1, \ldots, x_s) = x_{i+1} + a, \quad \text{where } x_i = a. \tag{59}$$

In our experiments, we set $\mathcal{A} = \{1, 2, 3, 4\}$, $\mathcal{Z} = \{5, \ldots, 100\}$, and $s = 10$. To introduce synonymy among anchors, we modify the mapping as

$$\mathcal{F}(x_1, \ldots, x_s) = x_{i+1} + (a \bmod 2), \quad \text{where } x_i = a, \tag{60}$$

so that anchors $\{1, 2\}$ and $\{3, 4\}$ produce equivalent outputs, mimicking synonymous relationships observed in natural language.

**Model and training hyperparameters.** Our model is a decoder-only Transformer with a single layer and a single attention head. The architecture follows the standard GPT design, consisting of a multi-head self-attention block and a position-wise feed-forward network. The Tanh activation function is used in the feed-forward network. A key aspect of our experimental setup is that both the token embedding layer and the final output projection layer are fixed and not updated during training. This allows us to isolate the learning dynamics exclusively within the Transformer's attention and feed-forward weights.

All trainable weights in the model are initialized from a normal distribution with a mean of 0. The standard deviation for different components is set based on the model dimension $d_{\text{model}}$ as $\sigma = d_{\text{model}}^{-0.85}$. The loss is computed only on the prediction of the last token in the sequence.

The model was trained for 30 epochs using the AdamW optimizer. We employed a learning rate scheduler that combines a gradual warmup phase for the first 10 epochs followed by a cosine annealing schedule. The specific hyperparameters are detailed in Table 1.

Table 1: Model and Training Hyperparameters

| Parameter | Value | Parameter | Value |
|---|---|---|---|
| Model Architecture | | Training Settings | |
| Vocabulary Size | 201 | Optimizer | AdamW |
| Model Dimension ($d_{\text{model}}$) | 640 | Batch Size | 1000 |
| Feed-Forward Dim. ($d_{\text{ff}}$) | 1280 | Epochs | 30 |
| Key/Value Dim. ($d_k, d_v$) | 640 | Weight Decay | 0.0 |
| Number of Layers | 1 | Gradient Clipping | 1.0 |
| Number of Heads | 1 | AdamW $\beta_1, \beta_2$ | 0.9, 0.999 |
| Activation Function | Tanh | | |
| *Learning Rate Scheduler (Warmup + CosineAnnealing)* | | | |
| Initial LR / $\eta_{\min}$ | $1 \times 10^{-5}$ | Warmup Epochs | 10 |
| Warmup Multiplier | 15.0 | Cosine Annealing $T_{\max}$ | 200 |

## B.2 Synthetic Dataset without activation function

In this section, we show similar result for model without activation function, as a supplement to the synthetic data experiments. This shows that it is reasonable to ignore activation in our analysis

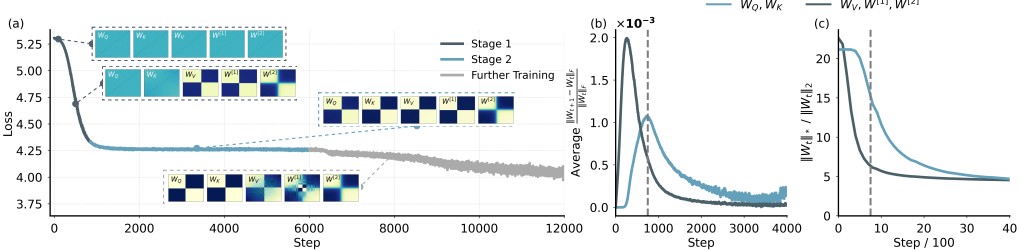

Figure 4: (a) Evolution of cosine similarity matrices for outer and attention parameters. The training process is partitioned into three stages: Condensation (Stage 1), Key-Query rank collapse (Stage 2), and a further training stage. Stage transitions are identified by plateaus in the loss curve and structural shifts in these matrices. (b) The relative change of norms between attention and outer parameters. The gray dashed line marks the onset of Stage 2, where updates to the attention parameters begin to dominate. (c) Evolution of the effective rank for both parameter groups, tracking the change in their intrinsic dimensionality throughout training.

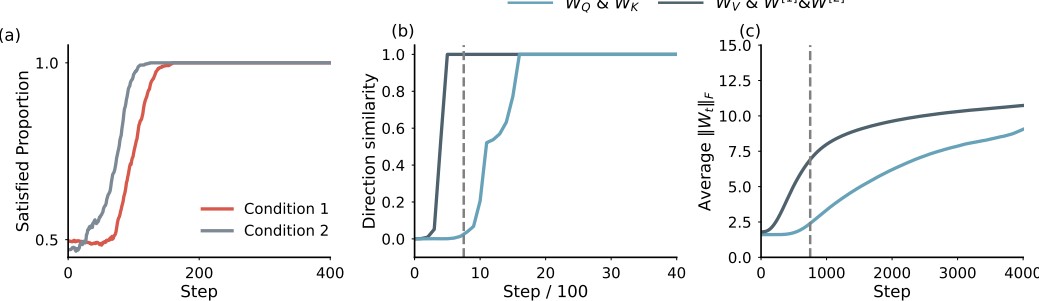

Figure 5: (a) Proportion of satisfied conditions in Assumption 1, measured as $\frac{|A_1|}{d_m}$ and $\frac{|A_2|}{d_m^2}$. (b) Similarity between singular vectors of two adjacent time steps. (c) Frobenius norms of parameter groups.

### B.3 Experimental Setting of Real Task

To validate that our theoretical insights generalize beyond simplified settings, we conducted experiments on the WikiText dataset, a standard benchmark for language modeling. This experimental setup intentionally incorporates more complex and commonly used architectural features.

**Dataset and Task.** We use the WikiText dataset, which consists of high-quality articles from Wikipedia. The task is next-token prediction, where the model is trained to predict the next word in a sequence. Consistent with our synthetic experiments, the training objective is calculated exclusively based on the prediction loss for the final token of each input sequence. The sequence length is set to 2048.

**Model and Training Hyperparameters.** We use a 2-layer decoder-only Transformer. To test the robustness of our findings, this model's architecture includes standard components that were abstracted away in the synthetic setup. Specifically, it incorporates residual connections after both the self-attention and feed-forward sub-layers, and it utilizes the GeLU activation function in the feed-forward network. This more realistic configuration allows us to demonstrate that our theory holds even in the presence of such non-linearities and standard architectural features.

All model weights are initialized from a normal distribution with a standard deviation of $\sigma = d_{\text{model}}^{-1.2}$. The model was trained for 5 epochs using the AdamW optimizer with an initial learning rate of $2 \times 10^{-4}$, which was managed by a cosine decay schedule with a warmup phase. The detailed hyperparameters for this experiment are listed in Table 2.

Table 2: Model and Training Hyperparameters for WikiText

| Parameter | Value | Parameter | Value |
|---|---|---|---|
| Model Architecture | | Training Settings | |
| Vocabulary Size | 31,999 | Dataset | WikiText |
| Model Dimension ($d_{\text{model}}$) | 64 | Sequence Length | 2048 |
| Feed-Forward Dim. ($d_{\text{ff}}$) | 800 | Optimizer | AdamW |
| Key/Value Dim. ($d_k, d_v$) | 64 | Batch Size | 500 |
| Number of Layers | 2 | Epochs | 5 |
| Number of Heads | 1 | Learning Rate | $2 \times 10^{-4}$ |
| Activation Function | GeLU | AdamW $\beta_1, \beta_2$ | 0.9, 0.999 |
| Pos. Emb. Length | 2048 | Weight Decay | 0.0 |
| | | Gradient Clipping | 1.0 |

## C  Experiments Compute Resources

The experiments were conducted on a server with the following configuration:

- 48 AMD EPYC 7352 24-Core Processors, each with 512KB of cache

- 251GB of total system memory

- 8 NVIDIA GeForce RTX 4080 GPUs with 16GB of video memory each

- The experiments were run using Ubuntu 22.04 LTS operating system

$\square$

