# OpenReview forum: "From Condensation to Rank Collapse: A Two-Stage Analysis of Transformer Training Dynamics"
_NeurIPS.cc/2025/Conference — NeurIPS 2025 oral_

### Official Review · Reviewer_t9Wk · 2025-06-27

**Clarity:** 2
**Significance:** 3
**Originality:** 2
**Rating:** 4
**Confidence:** 2

**Summary:**

The authors present a theoretical analysis of linearized transformer training dynamics under small initialization using a gradient flow framework. They identify a two-stage learning process: first, outer-layer weights (MLP, value matrices) escape the small initialization regime and undergo a directional alignment process called condensation. Then, the key and query matrices become active and exhibit rank collapse. The analysis is formalized for a one-layer transformer trained on a binary classification task. The authors prove that condensation and rank collapse emerge generically from non-degenerate random initialization. They support their theory with a synthetic experiment that shows the evolution of the weight matrices during training for a single task.

**Questions:**

1. See W1.
2. In L.28 you say that modern models are extremely overparameterized. Can you give a reference that underscores this points for language models?
3. Same for using dropout in language models.
4. There are some non-standard terms that are used without clear definition or citation, e.g. rank-collapse (L.58), dichotomy assumptions (L.53), could you define them?
5. The related work part explains other work in dynamics, but it would be useful to include further details on how the work mentioned relates to your contributions, other than also being about dynamics. For example, it is unclear to me why ICL is mentioned so much, but I might be missing the connection. In addition L.106 should not go into the Related Work section, as you are defining new novel terms.
6. Paragraph L.113 I have a hard time understanding. How do the assymetries come into play that make your definition advantageous. Especially as later on you introduce some assumptions on dynamical symmetries?
7. This is probably my missing background, but what are the assumptions you need to make on X and y for binary classification? Do they need to scale in a certain way? Can it be any task?
8. Where is the dot{w} in L.218/219 defined?
9. L.175 claims the analysis can easily be extended to other loss functions. All loss functions? Only logistic loss? Does (7) change then?
10. You omit the activation function in L.296 for the second part of the analysis. Why is this ok? Does it change the dynamics in your example if you leave it out?
11. Where does the index in $\mathcal L$ come from in (15)?
12. In the experimental section, what was the criterion for outlining phases 1 and 2? Since you consider the blow up in weights, a less quantitative approach could be to measure precisely those quantities. in addition, it would be great to see the weights before and after the given phase, to see if the change is then aprupt or not.
13. Your experiment is executed once. How robust is the timing of the stage, and the analysis you have for different seeds?

Small things
- You could consider using commands to cite that do not include the author names when not directly used in the text. Right now it is slightly difficult to parse at times because it is so bloated.
- L37 you use some parameters without truly defining them, consider refering to the definition and naming them outer/inner parameters already here (L.57). Without the context what follows this is difficult to understand.

**Ethical Concerns:**

["NO or VERY MINOR ethics concerns only"]

**Final Justification:**

I updated my score for borderline acceptance (see my reply to the authors). But probably I was not the ideal person to do the review, hence my hesitation.

**Limitations:**

I understand that linearlization is a limited setting, but still worth exploring on its own right. However, it is unclear to see how the assumptions that happen before training phase 2 would translate to larger networks and more complex tasks, I have a hard time imagining and analogy which would result in a deeper motivation (small initializations help big models, but do they follow two stage learning?).

**Quality:**

2

**Strengths And Weaknesses:**

S1 The authors analyse a linearized transformer’s dynamics for small initializations, showing that there are two phases, which is an advance over previous work.

S2 The idea of condensating amtrices seems to be interesting, but could be motivated a bit more clearly.

S3 The two phases are shown in an example task.

W1 In the introduction L.22, the authors claim that it is desirable to have dynamical analysis that holds independently of the specific task. However, I find it difficult to understand where in the manuscript this question is adressed, e.g. the task in section 4 only describes a single instance, and the assumptions on the input data in the theoretical analysis are not spelled out in the main.

---

> ### Author Rebuttal · Authors · 2025-07-30
>
> We sincerely thank you for your detailed and insightful feedback, which helps us significantly improve the clarity and rigor of our paper. We address each of your points below.
>
> ## Response to Question 1
> Our theoretical analysis does not rely on strong data assumptions—only the mild condition
>
> \begin{equation*}
>     \boldsymbol{v} := \frac{\sum_{i=1}^n y_i \left(\sum_{j=1}^s X_{i,j}\right)}{\left\| \sum_{i=1}^n y_i \left(\sum_{j=1}^s X_{i,j}\right) \right\|_2} \neq 0.
> \end{equation*}
> which holds in very general settings. Thus, the analysis is largely task-independent. To empirically demonstrate this generality beyond our synthetic task, we conducted new experiments on the Wikitext-103 dataset using a standard 2-layer Transformer with GeLU activation and residual connections. To visualize the result, we define the following aggregation index to reflect our results. According to the definition 1 of the article. A neuron is condensed if its alignment with the dominant direction exceeds:
> $$
>  \left| \left\langle \frac{\boldsymbol{W}\_k}{\| \boldsymbol{W}\_k \|}, \boldsymbol{v} \right\rangle \right| \ge 0.95
> $$
> The condensation rate is defined as follows:
> $$
> \text{Condensation Rate} = \frac{\text{\\# condensed neurons}}{\text{total neurons}}
> $$
>
> We plot the condensation rate of the first layer of decoder.
>
> | Condensation Rate / Epochs | 0.025 | 0.075 | 0.125 | 0.15 | 0.2 |
> |-------------------------|-------|-------|-------|------|------|
> | $\boldsymbol{W}\_Q$ | 0.11 | 0.11  | 0.77  | 0.99 | 0.99  |
> | $\boldsymbol{W}\_V$ | 0.11 | 0.9  | 0.91  | 0.84 | 0.76  |
> | $\boldsymbol{W}^{[1]}$ | 0.10  | 0.91  | 0.97  | 0.98 | 0.96  |
>
> Our findings on this 2-layer Transformer confirm that the same two-stage learning dynamics emerge, directly addressing the concern about whether our results generalize to larger networks and more complex tasks. Specifically, we observed that the $W_V$ and MLP weights condense first, followed by the $W_Q$ and $W_K$ matrices, consistent with our theory.
>
>
> ## Response to Question 2
> By "extremely overparameterized," we refer to the empirical observation that modern LLMs have far more parameters than needed to fit their training data. This is supported by scaling laws, as shown by Kaplan et al. [1, Section 3.2], where larger models consistently yield better performance despite having orders of magnitude more parameters than data points.
>
> ## Response to Question 3
> Dropout remains an important regularization technique in modern language models. For example, the Qwen-7B/72B technical reports [3] explicitly state in Section 2.3 that a dropout rate of 0.1 is consistently applied after both attention and feed-forward layers.
>
> ## Response to Question 4
> Rank-collapse refers to the degeneration of parameter or attention matrices into low-rank states. In our work, this specifically describes the collapse of the $W_Q$ and $W_K$ matrices. Similar phenomena have been discussed in prior works such as [2].
> Regarding the dichotomy assumptions, its absence means that Theorem 1 holds almost everywhere, rather than implying the dynamics necessarily degenerates or diverges. In other words, Theorem 1’s conclusions apply broadly without requiring this assumption.
>
> ## Response to Question 5
> We emphasize in-context learning (ICL) because recent studies in ICL provide well-developed dynamical frameworks under the sequential prediction paradigm, which inspire task-agnostic analysis. This perspective informs our approach and allows our dynamical analysis to remain independent of specific tasks, which we view as a key contribution. We agree that Line 106 introduces novel terminology and will relocate it outside the Related Work section in the revision.
>
> ## Response to Question 6
> In the linearized model, our dynamics (10) exhibits a high degree of symmetry, which induces useful conservation laws. However, purely symmetric setups can lead to degeneracies—for example, certain measurements may vanish due to exact cancellations. To avoid this, we rely on initial asymmetries to break such degeneracies and make the conserved quantities informative. As training progresses, the dynamics tend to restore symmetry through parameter evolution. This motivates our introduction of the cohesion condition to ensure that the system evolves toward a symmetric structure in a controlled way.
>
> ## Response to Question 7
> As mentioned in our response to Question 1, we do not require any specific assumptions on the dataset. Our analysis is general and can be applied to various tasks, including binary classification, without constraints on the scaling of $X$ and $y$. This flexibility is one of the strengths of our approach.
>
> ## Response to Question 8
> The notation $\dot{\boldsymbol{W}}$ is shorthand for the time derivative of the matrix, i.e., $\frac{\mathrm{d}}{\mathrm{d} t} \boldsymbol{W}$. We will clarify this in the manuscript to avoid confusion.
>
> ## Response to Question 9
> Our analysis can be extended to all loss functions for which the condition
> $$
> l'(0) < 0.
> $$
> holds. In such cases, equation (7) will remain unchanged.
>
> ## Response to Question 10
> Although we omit the activation function in the second part of the analysis, this is acceptable because, at this stage, the entries for $W_V$, $W_1$, and $W_2$ remain small. For activation functions like $\tanh$, the absence of a second-order term in its Taylor expansion means the model remains approximately linear with respect to $W_V$, $W_1$, and $W_2$. Therefore, omitting the activation function simplifies the analysis without affecting the dynamics, as the higher-order terms of $\tanh$ are negligible during this stage.
>
> To further address the reviewer's concern, we conducted additional experiments by removing the activation function on synthetic dataset. The condensation rate, defined as the ratio of condensed neurons to total neurons, showed that the dynamics remained consistent with the original experiment, with condensation occurring at similar time points. The minor misalignment is due to adjustments made to the initialization for training stability.
>
> | Condensation Rate / Epochs | 0 | 15 | 20 | 25 |
> |-------------------------|-------|-------|-------|------|
> | $W_Q$ | 0.04 | 0.04  |  0.05  | 0.84 |  0.97 |
> | $W^{[2]}$ | 0.02  | 0.83  |  0.85  |  0.81 | 0.83  |
>
> ## Response to Question 11
> The index in equation (15) corresponds to the sample index, which is defined in Section 3 of the paper.
>
> ## Response to Question 12
> Thank you for your insightful comment. We define the first and second stages based on the degree of parameter aggregation and the change of norm. Specifically, the transition between stages is marked by the following:
> - In the first stage, we observe minimal change in the Frobenius norm of $W_Q$, while the norm of $W^{[2]}$ increases significantly.
> - The second stage begins when the norm of $W_Q$ starts to increase, which coincides with a more substantial change in model behavior.
> To visualize this, we provide the Frobenius norms of $W_Q$ and $W^{[2]}$ at various epochs in the table below. As shown, the norm of $W_Q$ remains static until epoch 13, where it begins to grow abruptly by an order of magnitude, marking a clear and sudden transition to the second phase.
>
> | Frobenius Norm / Epoch         | 0     | 13 | 15 | 19 | 25 |
> |-------------------------|-------|-------|-------|------|------|
> | $W_Q$ | 1.1 | 1.1  | 11.0  | 19.4 | 47  |
> | $W^{[2]}$ | 3.5  | 12.8  | 13.2  | 18.8 | 20.0  |
>
> ## Response to Question 13
> To evaluate the robustness of the stage timing and the analysis across different seeds, we've conducted five additional runs with seeds 1843, 1984, 2000, 2025, and 2048. The timing of the stage transitions has been measured by tracking the condensation rate of $W_Q$. The results demonstrate a high consistency in the transition between stages, even across different initializations.
>
> The table below shows the condensation rates for  $W_Q$ at different epochs, with the corresponding mean and standard deviation:
> | Condensation Rate / Epoch         | 10 | 11 | 12 | 13 | 14 |
> |-------------------------|-------|-------|-------|------|------|
> | $W_Q$ | $0.04 \pm 0.0004 $  | $0.05 \pm 0.007$  | $0.38 \pm 0.26$ | $0.87 \pm 0.03$ | $0.93 \pm 0.01$  |
>
> As seen in the table, the condensation rate shows a clear progression across epochs, with minimal variation between runs. The standard deviations indicate that the transitions are stable, and there is a high level of reproducibility in the dynamics of the system for different seeds.
>
> This consistency reinforces the robustness of the stage timing in the training process.
>
>
> ## Reference
> [1] Kaplan, Jared, et al. "Scaling laws for neural language models." arXiv preprint arXiv:2001.08361 (2020).
>
> [2] Noci, Lorenzo, et al. "Signal propagation in transformers: Theoretical perspectives and the role of rank collapse." Advances in Neural Information Processing Systems 35 (2022): 27198-27211.
>
> [3] Bai, Jinze, et al. "Qwen technical report." arXiv preprint arXiv:2309.16609 (2023).

---

> > ### Comment · Reviewer_t9Wk · 2025-08-05
> >
> > Dear authors,
> >
> > Thank you for your reply, which I have considered along with the other reviewers reports and replies.
> > Your clarifications improved my understanding of your work - and while I do not fully understand the details of the analysis I can now grasp its use better.
> >
> > I also apprechiate additional experiments in the setting you already have, including extra experiments in a more general setting where you demonstrate condensation.
> >
> > Best!

---

> > > ### Author Response · Authors · 2025-08-07
> > >
> > > We would like to extend our sincere gratitude for your thorough review and perceptive comments. The manuscript has been substantially improved as a direct result of your feedback, and we would be pleased to address any further points you may have.

---

> ### Author Response · Authors · 2025-08-04
>
> Dear Reviewer t9Wk,
>
> We hope this message finds you well! Thank you for your insightful review of our paper. As for the concerns about our work in the review, we have provided very specific and detailed responses. Have these responses resolved your concerns? If you have any further questions about our work or responses during this discussion period, please do not hesitate to contact us. We sincerely look forward to receiving your further comments on our responses.
>
> Once again, thank you for your valuable time and feedback!
>
> Best regards,
>
> Authors

---

### Official Review · Reviewer_DNga · 2025-06-30

**Clarity:** 3
**Significance:** 3
**Originality:** 3
**Rating:** 5
**Confidence:** 4

**Summary:**

The authors provide an in depth analysis on the training dynamics of a single-layer transformer model with small initialization trained on a binary classification task with exponential loss. Their theoretical contributions can be separated into three key parts/stages:
1. By analyzing the linearization of the loss, they show that the loss is initially guided by the gradient of an energy functional that depends only on W^V, W^[1], and W^[2]. This first shows that the initial stages of optimization optimize only these three matrices while leaving W^Q, and W^K primarily static. They additionally show that this energy functional diverges to infinity in finite time (almost surely) and use this as a means to show that the norms of the three matrices are blowing up and capable of escaping the small initialization.
2. Using the dynamics dictated in the previous part, they use this to establish a condensation condition under which W^V experiences condensation.
3. Lastly, the authors leverage the dynamics to motivate assumptions that would lead to rank collapse in the matrices W^Q, and W^K. Notably, these assumptions assert that the second stage in training emerges after the W^V, W^[1], and W^[2] have been sufficiently optimized where the W^Q and W^K have also shrunk in norm (Key-Query stunting). During this second stage, the W^V, W^[1], and W^[2] are assumed to be constant and the W^Q, W^K matrices evolve according to a linear ODE. The extent of the rank-collapse hence depends on the multiplicity of the top singular value of the linear ODE.

**Questions:**

- The experimental setup is also lacking a bit of clarity. What is the objective? Is it to approximate the anchor function? If so, why would it make sense to use the cross-entropy loss?
- For the experiments, it is unclear to me what the cosine similarity is taken over. The caption says that it’s taken across the neural network layers but the model is only a single layer. It would be great if an equation is added describing exactly what the visualizations are showing.
- Does Assumption 2 hold in the experiments provided in Section 4? Would it not be possible to measure and show that the key-query stunting and that the criticality conditions hold?
- Similarly, with Theorem 3, could a soft measure of rank be used to show that rank collapse emerges with training in the experimental setting provided in Section 4?
- Could the measurements in Section 4 be performed on at scale transformers to show that the stages emerge in actual training and that the results are not just limited to the synthetic setting?

**Ethical Concerns:**

["NO or VERY MINOR ethics concerns only"]

**Final Justification:**

The authors have addressed all the concerns I raised, and I have raised the score to accept in response.

**Limitations:**

Both theoretical and empirical analysis is performed in synthetic settings. While the theoretical limitations is adequately addressed in their Limitations section and reasonable, the paper would strongly benefit including additional at-scale empirical experiments that would provide evidence for the behaviour that their paper is showing.

**Quality:**

3

**Strengths And Weaknesses:**

Strengths:
- Provides a solid theoretical analysis on the training dynamics of a single layer transformer model that could be used to explain why rank collapse emerges in practise.
- Separates and groups the training of a transformer block into two different stages which is backed by both empirical experiments and a theoretical derivation.

Weaknesses:
- $m$ is not defined prior to definition 1.
- W^[1] and W^[2] are being referenced prior to being properly defined and cannot be as easily deduced as W^Q, W^K, W^V.
- Structurally, the placement of the experimental results (Section 4) seems out of place to me. There is little to no motivation provided prior to the presentation of the experiments so it is unclear to me what the purpose of the experiments are when they are introduced. It is only until after the theoretical exposition does it become clear what the experiments are trying to show (i.e. the two different stages that are governed by the training dynamics).
- Section 5.3 is referencing experiments/empirical evidence that don’t seem to be included in the paper.
- All of the work done in the paper (including empirical results) are performed on miniature/synthetic settings and additionally, many of the theoretical assumptions that can be verified empirically in the synthetic setting in Section 4 are not verified empirically. While it is reasonable to perform the theoretical analysis in the synthetic setting, the paper would benefit by including empirical evidence that the behaviour dictated by their training dynamics emerge in larger models/in practise as well. Currently, it is unclear if any of the results in the paper actually generalize.

---

> ### Author Rebuttal · Authors · 2025-07-30
>
> We would like to express our sincerest gratitude for your thorough and highly constructive review.
>
> We were very encouraged by your positive assessment of our paper's quality, clarity, significance, and originality. We respectfully ask if you might be willing to reconsider the negative overall evaluation, in light of your positive scores on these individual aspects. We hope that following detailed responses have fully addressed your concerns and now better align the paper's overall contribution with the strengths you identified.
>
> ## Response to Weaknesses
> We sincerely appreciate these structural observations and will implement the following revisions:
>
> > $m$ is not defined prior to definition 1.
>
> We will reposition ​​Definition 1​​ to immediately follow Section 3.1.
>
> > $W^{[1]}$ and $W^{[2]}$ are being referenced prior to being properly defined and cannot be as easily deduced as $W\_Q$, $W\_K$ and $W_V$.
>
> We will adjust the statement in L37 to  “We delineate different training dynamics for parameter matrices in Transformers.” and adjust the statement in section 4.
>
> > The placement of the experimental results (Section 4) is inappropriate.
>
> We will relocate experiments to follow Section 5 (Theoretical Analysis) and add new introductory paragraph:  "We now empirically validate our theoretical findings regarding stage-dependent dynamics (Section 5). These experiments specifically test two predictions: (1) initial condensation phenomenon, and (2) bifurcation of training dynamics into distinct stages."
>
> > Section 5.3 is referencing experiments/empirical evidence that don’t seem to be included in the paper.
>
> We apologize for the lack of experimental evidence and add the following:
>
> ### Empirical Validation of Assumption 1
>
> **Empirical Validation Framework**
>
> We quantitatively evaluate both conditions in Assumption 1 using custom-defined satisfaction metrics which reflects how many neurons satisfy the hypothesis:
>
>    - **Assumption 1.1 satisfied rate** = $\frac{|A\_1|}{d\_m}$ where
>      $A\_1 = \left \\{ i \in [d\_m] \mid \boldsymbol{W}^{[2]}\_i \boldsymbol{W}\_{\mathbf{v}} \boldsymbol{W}^{[1],i} >0 \right\\}$
>    - **Assumption 1.2 satisfied rate** = $\frac{|A\_2|}{d\_m^2}$ where
>      $A\_2 = \left\\{ (i,j) \in [d\_m] \\times [d\_m] \mid \langle \boldsymbol{W}^{[2]}\_i \boldsymbol{W}^{[1],i}, \boldsymbol{W}^{[2]}\_j \boldsymbol{W}^{[1],j} \rangle > 0 \right\\}$
>
> #### 2. Key Results: Rapid Convergence to Full Satisfaction
> Quantitative measurements across initial training epochs reveal:
>
> | Metric / Epoch         | 0     | 0.022 | 0.056 | 0.09 | >0.3 |
> |-------------------------|-------|-------|-------|------|------|
> | Assumption 1.1 satisfied rate | 0.525 | 0.52  | 0.65  | 0.95 | 1.0  |
> | Assumption 1.2 satisfied rate | 0.50  | 0.51  | 0.70  | 0.95 | 1.0  |
>
> The satisfaction rates start near 50% at initialization, as expected for random vectors. They then rapidly converge, surpassing 95% by 0.1 epochs and reaching a stable 100% by 0.3 epochs. This state of full satisfaction persists throughout the condensation phase we analyze.
>
> ### Conclusion
> This empirical evidence confirms that Assumption 1 is not merely a theoretical convenience but a condition that is reliably and quickly met in practice. This validates the foundational premise of our condensation analysis, directly addressing the reviewer's concern.
>
> > The lack of theoretical hypothesis verification and larger model experiments.
>
> We will verify the theoretical hypothesis and provide experimental evidence for a larger transformer.
>
> ### 1.Validation of Dynamics Separation Assumptions
>
> We provide direct empirical validation of Assumption 2 through quantitative measurement of parameter dynamics during training. Our analysis reveals clear separation behavior consistent with the theoretical assumptions.
>
> **Empirical Validation Method**
>
> We quantify the dynamical separation phenomenon using the **relative change rate** metric:
> $$
> \text{relative change rate} = \frac{\\|\boldsymbol{W}\_{t+1} - \boldsymbol{W}\_{t}\\|}{\\|\boldsymbol{W}\_{t}\\|}
> $$
> This measures the relative magnitude of parameter updates between training steps.
>
> **Key Results: Separation at Epoch 13**
>
> Measurements at the critical transition point (epoch 13) demonstrate:
>
> | relative change rate / epoch | 0 | 2 | 10 | 13 | 14 | 16 | 25 |
> |-|-|-|-|-|-|-|-|
> | $W_Q$ | 8.3e-06  | 1.3e-06  | 9.4e-07  | 8.2e-03  | 5.0e-04  | 1.4e-06  | 6.8e-05  |
> | $W_K$ | 8.0e-06  | 1.2e-06  | 1.0e-06  | 7.8e-03  | 4.3e-04  | 8.5e-07  | 1.2e-05  |
> | $W_V$ | 1.2e-03  | 3.3e-05  | 1.4e-04  | 5.3e-05  | 7.5e-04  | 1.9e-06  | 5.5e-05  |
> | $W^{[1]}$ | 1.1e-03  | 3.1e-05  | 2.5e-04  | 6.4e-05  | 1.0e-04  | 6.7e-05  | 5.3e-05  |
> | $W^{[2]}$| 1.9e-03  | 3.3e-04  | 1.1e-04  | 9.6e-05  | 8.5e-05  | 4.0e-05  | 4.5e-05  |
>
> **Critical observations**:
> 1. **Attention parameters ($\boldsymbol{W}_Q,\boldsymbol{W}_K$)** exhibit large update magnitudes $\approx 1e-2$
> 2. **Linear path parameters ($\boldsymbol{W}_V,\boldsymbol{W}^{[1]},\boldsymbol{W}^{[2]}$)** show minimal updates ($\le 1e-4$)
>
> This empirical evidence directly supports Assumption 2 which validates the core claim that $\boldsymbol{W}_Q,\boldsymbol{W}_K$ evolve while $\boldsymbol{W}_V,\boldsymbol{W}^{[1]},\boldsymbol{W}^{[2]}$ remain quasi-static.
>
> ### 2. Larger transformer experiments
>
> We confirm the ​​universality of condensation dynamics​​ through new experiments on ​​Wikitext-103​​ (natural language corpus) using a standard 2-layer decoder-only transformer with GeLU activation and residual connections. This demonstrates the phenomenon’s independence from specific data distributions.
>
> ### Experimental Results
>
> **Condensation Rate**:
>
> We define the following aggregation index to reflect our results. According to the definition 1 of the article. A neuron is condensed if its alignment with the dominant direction exceeds:
> $$
>  \left| \left\langle \frac{\boldsymbol{W}\_k}{\| \boldsymbol{W}\_k \|}, \boldsymbol{v} \right\rangle \right| \ge 0.95
> $$
> The condensation rate is defined as the number of condensed neurons divided by the total number of neurons.
>
> **Key Result**
> We plot the condensation rate of the first layer of decoder.
>
> | Condensation Rate / Epochs | 0.025 | 0.075 | 0.125 | 0.15 | 0.2 |
> |-------------------------|-------|-------|-------|------|------|
> | $\boldsymbol{W}\_Q$ | 0.11 | 0.11  | 0.77  | 0.99 | 0.99  |
> | $\boldsymbol{W}\_V$ | 0.11 | 0.9  | 0.91  | 0.84 | 0.76  |
> | $\boldsymbol{W}^{[1]}$ | 0.10  | 0.91  | 0.97  | 0.98 | 0.96  |
>
> Experimental results show that $\boldsymbol{W}\_V$ $W^{[1]}$ parameters converge quickly (at epoch 0.075), while $\boldsymbol{W}\_Q$ converges significantly later than the former (at epoch 0.15). This shows the applicability of our results on real language tasks.
>
> ## Response to Question 1
>
> Yes, the anchor function is formulated as a classification task, following common practice in prior works (e.g., [arxiv:2306.00802, 2401.08309]). We use cross-entropy loss to align with standard transformer training and better approximate downstream language tasks.
>
> ## Response to Question 2
> Thank you for pointing this out, and we apologize for the confusion caused by the caption. You are correct that the model consists of a single layer. Our intention was not to present cosine similarities across layers, but rather to show the cosine similarities of different parameter matrices.
>
> We clarify the computation of cosine similarity of the prarameter matrix. Given a parameter matrix $\boldsymbol{W} \in \mathbb{R}^{n \times d}$, we compute the cosine similarity matrix as follows:
>
> - Normalize each row: $\tilde{\boldsymbol{W}}_i = \frac{\boldsymbol{W}_i}{|\boldsymbol{W}_i|_2}$
>
> - Compute the cosine similarity matrix: $\tilde{\boldsymbol{W}} \tilde{\boldsymbol{W}}^\top \in \mathbb{R}^{n \times n}$
>
> This results in a matrix where each entry represents the cosine similarity between a pair of rows in $\boldsymbol{W}$, i.e., between two parameter vectors.
>
> ## Response to Question 3
>
> Yes, Assumption 2 is verified in our experiments. As shown in our response to Weakness 5, the key-query stunting and criticality conditions are satisfied around epoch 13.
>
> ## Response to Question 4
> Yes, we address this by introducing the concept of effective rank as a soft measure of matrix rank. Specifically, for a parameter matrix $\boldsymbol{M}$ with singular values $\sigma_1 \ge \sigma_2 \ge \dots \ge \sigma_n$, we define:
> $$
> \text{erank}(\boldsymbol{M}) = \sum_{i=1}^n \frac{\sigma_i}{\sigma_1}
> $$
> This quantity captures how concentrated the spectrum is, with lower values indicating stronger rank collapse. We compute the effective rank of several parameter matrices across different training epochs, as shown below:
> | Effective rank / Epochs | 0.025 | 0.075 | 0.125 | 0.15 | 0.2 |
> |-|-|-|-|-|-|
> | $W_Q$ | 6.8 | 6.8  | 3.36  | 1.28 | 1.26  |
> | $W_V$ | 6.8 | 2.54  | 1.92  | 2.22 | 2.28  |
> | $W^{[1]}$ | 7.9  | 3.38  | 2.38  | 1.94 | 1.98  |
>
> We observe a sharp drop in effective rank beginning around epoch 13, which corresponds to the second stage of training. This provides clear evidence that rank collapse emerges during this phase, in line with the theoretical predictions of Theorem 3.
>
> ## Response to Question 5
>
> We have included additional experiments on the WikiText dataset, which show that the same training stages also emerge in real-world settings.

---

### Official Review · Reviewer_7v7g · 2025-07-01

**Clarity:** 2
**Significance:** 2
**Originality:** 3
**Rating:** 5
**Confidence:** 4

**Summary:**

This paper aims to rigorously investigate gradient flow dynamics of Transformers with small initialization via dynamics of individual weight matrices in the model. Their main results are based on the ‘condensation’ phenomenon wherein the rows and columns of weight matrices converge in direction (upto sign) to the same vector, in stages starting with W_V and MLP matrices followed by Key, Query matrices. This phenomenon is empirically demonstrated via training on a toy dataset.

**Questions:**

1. To understand if condensation occurs more generally, is it possible to show that the same empirical results hold (for any of the below)

- for Transformers with residual connections,
- beyond the toy dataset used,
- for MLP activation functions other than tanh like ReLU/GELU, and
- for Transformers with more than one layer?

2. Section 4:
- What is the exact initialization used for this experiment?
- I could not understand the motivation for the constructed dataset: could the authors elaborate what language task it is simulating?
- Could the authors clarify how the cosine similarity matrices of Fig 1 are calculated? Also, what are the numerical values on y-axis in Fig 1?
- It looks like $W^{[1]}$ is no longer in the condensation state at the end of training in Fig 1 - is this expected?
- Is there a chance that training for longer time beyond the epochs reported leads to a sudden change in the model, somewhat similar to the grokking phenomenon [1]?

3. Eq 10/ Line 193 : The product $vW_V$ cannot be computed since $v$ is a vector; did the authors mean $v^\top W_V$? Also, by definition 1 $W_V \in \mathbb{R}^{d_m \times d_m}$, while $v \in \mathbb{R}^d$ from Eq. (8); is $d=d_m$?

4. Line 289: "matrices $W_Q$ and $W_K$ demonstrate remarkable stability in scale" - is it possible to show that this holds empirically? (since this does not follow from cosine similarity maps in Figure 1)

**Ethical Concerns:**

["NO or VERY MINOR ethics concerns only"]

**Final Justification:**

My main concern about this work were limited applicability beyond the toy experimental setup in the original version. The authors have addressed that via experiments on wikitext-103 data with a decoder-only Transformer with GELU activation and residual connection. They have also clarified some questions I had about measuring cosine similarity / experimental setup etc. and a question on experimentally verifying stability in scales of weight matrices. Hence I've raised my score to 5.

**Limitations:**

Yes

**Quality:**

2

**Strengths And Weaknesses:**

Strengths:

Rigorously analyzes stagewise optimization dynamics of Transformer using weight matrix dynamics, supported by experiments on a simple dataset and model.

Weaknesses:

1. The data setup seems to be very specific, and it is not clear if it is applicable to natural language data (or any other dataset except the one in the paper). Unless condensation is consistently observed for at least some other datasets, it is difficult to assess its general applicability.

2. The Transformer architecture is unconventional: there are no residual connections for the Attention / MLP output, and activation function for MLPs is tanh instead of the commonly used ReLU/GELU.

---

> ### Author Rebuttal · Authors · 2025-07-30
>
> We sincerely appreciate your insightful comments, which have greatly contributed to improving the quality of our work. Below is a point-by-point response to your feedback.
>
> ## Weakness 1
> >"Is condensation consistently observed beyond synthetic datasets?"
>
> **Response:**
> We confirm the ​​universality of condensation dynamics​​ through new experiments on ​​Wikitext-103​​ (natural language corpus) using a standard 2-layer decoder-only transformer with GeLU activation and residual connections. This demonstrates the phenomenon’s independence from specific data distributions.
>
> ### Experimental Results
>
> 1. Condensation Rate
>
> We define the following aggregation index to reflect our results. According to the definition 1 of the article. A neuron is condensed if its alignment with the dominant direction exceeds:
> $$
>  \left| \left\langle \frac{{W}\_k}{\| {W}\_k \|}, {v} \right\rangle \right| \ge 0.95
> $$
> The condensation rate is defined as the number of condensed neurons divided by the total number of neurons.
>
> 2. Key Result
> We plot the condensation rate of the first layer of decoder.
>
> |Condensation Rate/Epochs|0.025|0.075|0.125|0.15|0.2|
> |-|-|-|-|-|-|
> |$W\_Q$|0.11|0.11|0.77|0.99|0.99|
> |$W\_V$|0.11|0.9|0.91|0.84|0.76|
> |$W^{[1]}$|0.1|0.91|0.97|0.98|0.96|
>
> On Wikitext-103, all key matrices rapidly condense (rates approach 1.0), confirming the phenomenon's generality. The observed temporal separation, where linear parameters condense earlier than attention parameters, also provides strong support for our multi-stage dynamics theory.
>
> ## Weakness 2
> >"Unconventional architecture: no residual connections, tanh instead of ReLU/GELU"
>
> **Response:** We address Weakness 2 from two perspectives: experimental validation and theoretical justification.
>
> ### 1. Experimental Setting
>
> As demonstrated in our response to Weakness 1, experiments using a 2-layer, 1-head decoder-only Transformer on Wikitext-103—which employs ​​GeLU activations​​ and ​​residual connections​​—confirm that our core findings generalize to architectures with modern activation functions and residual pathways. It shows the generality of our results to multi-layer gelu activation functions.
>
> ### 2. Theoretical Development
>
> **On the versatility of activation functions**
>
> Our theoretical results apply to all activation functions that satisfy the following conditions:
> $$
> \sigma(0) = 0 ~ \text{and} ~ \sigma'(0) \neq 0.
> $$
> This condition encompasses GeLU (validated experimentally) and excludes only functions with undefined or zero derivatives at the origin (e.g., ReLU).
>
> Stage 1 dynamics remain consistent​​ under this condition. The empirical loss expansion retains the same asymptotic form:
> $$
>     \mathcal{L} ({\theta})= \frac{1}{n} \sum\_{i=1}^n \left[ 1- \varepsilon^3 \left( \sum\_{j=1}^s \frac{1}{s} y\_i X\_{i,j} \bar{{W}}\_V \bar{{W}}^{[1]} \bar{{W}}^{[2]} \right) + o(\varepsilon^3) \right],
> $$
> yielding identical leading-order dynamics:
> $$
>     \frac{\mathrm{d} \bar{{\theta}}}{\mathrm{d} \bar{t}} = \nabla\_{\bar{{\theta}}} \left( {v}^{\intercal} \bar{{W}}\_V \bar{{W}}^{[1]} \bar{{W}}^{[2]} \right)
> $$
> So we get the same result for general activation function in the first stage of training.
>
> Stage 2 analysis​​ relies on the assumption that after Stage 1 convergence to a linear critical point, higher-order perturbations remain negligible. This holds for all σ satisfying the above condition, preserving our theoretical conclusions.
>
> **Discussion about residual connections**
>
> While residual connections warrant deeper exploration, we provide preliminary theoretical insights. Consider a single-layer Transformer with residuals:
> $$
> f\_{{\theta}} (X) = (X + Attn (X) + FFN(Attn(X))) W\_{\text{proj}}.
> $$
> The empirical loss expands as:
> \begin{equation*}
>     \mathcal{L} ({\theta})= \frac{1}{n} \sum\_{i=1}^n \left[ 1- \varepsilon \left(y\_i X\_{i,s} \bar{{W}}\_{proj} \right) + o(\varepsilon) \right].
> \end{equation*}
> Therefore, the first stage of analysis will be much simpler, since the leading order dynamics will be a linear ode governing ${W}\_{proj}$. Next, following the same principles as our second-stage analysis, we assume that the model reaches the critical point of the linear model dominated by $W\_{proj}$ and decompose the empirical loss as follows:
> $$
> \mathcal{L} ({\theta}) \approx \frac{1}{n} \sum\_{i=1}^n \mathcal{L}\_{1,i}({\theta}) \mathcal{L}\_{2,i} ({\theta}) \mathcal{L}\_{3,i} ({\theta})
> $$
> where
> $$
> \begin{aligned}
> \mathcal{L}\_{1,i} &= e^{-y\_i X\_{i,s} W\_{proj}} \\\\
> \mathcal{L}\_{2,i} &= e^{-y\_i Attn(X\_i)\_s W\_{proj}} \\\\
> \mathcal{L}\_{3,i} &= e^{-y\_i FFN(X\_{i,s}+Attn(X\_i)\_s) W\_{proj}}
> \end{aligned}
> $$
> By making reasonable assumptions, we can analyze the dominant dynamics at the moment. At this moment, the change of $W\_V$ will be most significant, rather than $W\_Q$ and $W\_K$. After all linear model changes stop, $W\_Q$ and $W\_K$ in the attention module begin to change. This ​​sequential learning phenomenon​​ aligns with the module-specific update ordering observed in our original work. Future work​​ will formalize this analysis for multi-layer architectures, but current results suggest residual connections preserve the core mechanistic principles we describe.
>
> ## Response to Question 1:
> To address the reviewer's concern about the generalizability of our findings on condensation, we provide the following clarifications. We have indeed confirmed that the condensation phenomenon holds across the various dimensions queried by the reviewer. These results are empirically and theoretically demonstrated in our responses to Weaknesses 1 and 2, as summarized below:
>
> - Residuals & Multi-Layer Models: Addressed by our extended theoretical analysis (Weakness 2) and empirically verified with a 2-layer model on Wikitext-103 (Weakness 1).
> - Alternative Activations (GELU/ReLU): Covered by our general theoretical proof and validated with GELU experiments (Weakness 1).
> - Beyond Toy Datasets: Confirmed via large-scale experiments on Wikitext-103 (Weakness 1, Table 1).
>
> ## Response to Question 2:
> We appreciate these technical inquiries and address them sequentially below.
>
> > What is the exact initialization used for this experiment?
>
> Given any trainable parameter matrix $W \in \mathbb{R}^{d\_1 \times d\_2}$, where $d\_1$ and $d\_2$ denote the input and output dimensions, respectively, its elements are initialized according to a normal distribution:
>
> $$
> W\_{i,j} \sim \mathcal{N}\left( 0,  d\_{1}^{-2\gamma} \right),
> $$
>
> where $\gamma$ is the initialization rate. Specifically, the initialization scale decreases as $\gamma$ increases. Note that $\gamma = 0.5$. In our experiments, we take $\gamma = 0.8$.
>
> > Could the authors elaborate what language task it is simulating?
>
> Our constructed dataset is designed to simulate algorithmic reasoning tasks, similar to those found in works like [arxiv:2306.00802, 2401.08309]. The primary goal is to create a controlled environment that isolates a model's ability to learn fundamental structural relationships between tokens, a core capability required for more complex language understanding.
>
> > How the cosine similarity matrices of Fig 1 are calculated? What are the numerical values on y-axis in Fig 1?
>
> - Calculation Method: For parameter matrix ${W} \in \mathbb{R}^{n \times d}$:
>   - Row-wise normalize: $\tilde{{W}}\_i = \frac{{W}\_i}{\\| {W}\_i\\|\_2}$
>   - Compute $\tilde{{W}} \tilde{{W}}^{\intercal}$
>
> - Y-axis Values: The numerical values on y-axis in Fig 1 is about 4-4.5 which corresponds to relative early stage of training.
>
> > It looks like $W^{[1]}$ is no longer in the condensation state at the end of training in Fig 1 - is this expected?
>
> Yes, this is entirely expected and a key part of our findings. Our results show that training proceeds in stages:
> - Stage 1 (epoch 0- 15): ${W}\_V$, ${W}^{[1]}$ and ${W}^{[2]}$ first condense into a low-complexity state to capture the task's simplest patterns.
> - Stage 2 (epoch 15-50): Other components, like ${W}\_Q$ and  ${W}\_K$, begin to condense.
> - Furthur Stage (epoch 50+) To achieve a lower loss and fit the task more accurately, the model must develop more complex and specialized representations. This requires the parameters to differentiate and move out of the condensed state, as observed for $W^{[1]}$.
>
> > Is there a chance that training for longer time beyond the epochs reported leads to a sudden change in the model, somewhat similar to the grokking phenomenon?
>
> This is an insightful question. While we did not observe the specific delayed generalization dynamic that defines grokking, the staged learning process we identify could be related. The transition from a condensed, low-complexity state to a differentiated, high-complexity one might be a foundational mechanism that precedes phenomena like grokking. We see this as a very promising avenue for future investigation.
>
> ## Response to Question 3:
> We confirm
>
> - Eq 10 contains a notation error. The correct term is ​${v}^{\intercal} \bar{{W}}\_V$.
> - Proposition 1 contains a notation error. The correct term is $X\_i \in \mathbb{R}^{s \times d\_m}$.
>
> ## Response to Question 4:
> This is an excellent point. We agree that cosine similarity does not capture matrix scale, and we appreciate the opportunity to provide direct empirical evidence.
>
> To demonstrate the different scaling dynamics, we report the Frobenius norm of $W\_Q$ and $W^{[2]}$ at key training epochs below.
>
> |Frobenius Norm/Epoch|0|13|15|19|25|
> |-|-|-|-|-|-|
> | $W_Q$ |1.1|1.1|11.0|19.4|47|
> | $W^{[2]}$ |3.5|12.8|13.2|18.8|20.0|
>
> As the data shows, the norm of $W\_Q$ is remarkably stable during the first training phase (e.g., from epoch 0 to 13). The subsequent sharp increase in its norm precisely marks the transition into the second learning phase we described. This behavior contrasts sharply with $W^{[2]}$ whose norm grows significantly within the first phase. This empirically confirms our claim that different model components exhibit distinct dynamic behaviors.

---

> > ### Comment · Reviewer_7v7g · 2025-08-01
> > **Thanks for your response**
> >
> > Thank you for your response and the additional experiments.
> >
> > My main concern about this work were limited applicability beyond the toy experimental setup in the original version. The authors have addressed that via experiments on wikitext-103 data with a decoder-only Transformer with GELU activation and residual connection. They have also clarified some questions I had about measuring cosine similarity / experimental setup etc. and a question on experimentally verifying stability in scales of weight matrices. Hence I've raised my score to 5.

---

> > > ### Author Response · Authors · 2025-08-02
> > >
> > > We sincerely thank you for your valuable engagement and insightful questions. Our paper is significantly more robust thanks to your guidance, and we welcome any further questions you may have.

---

### Official Review · Reviewer_S1TU · 2025-07-02

**Clarity:** 2
**Significance:** 3
**Originality:** 3
**Rating:** 4
**Confidence:** 4

**Summary:**

The paper investigates the training dynamics of transformers under a small initialization setting from the perspective of the gradient flow.  Specifically, the dynamics of attention modules are dissected into two stages: a) the first stage is called condensation stage, in which the core attention module (i.e., key-query matrices) is nearly stagnant (i.e., quasi-static) without significant fluctuations; b) the second stage, the quasi-static key-query matrices in the previous stage start to participate in training, leading to rank collapse. The paper provides rigorous analysis from the gradient flow to interprete the blow up dynamics and the condensation dynamics, and also show a so-called "dynamics separation" dynamics in the second stage.

**Questions:**

1. While the paper provides a rigorous and lengthy analysis for the so-called blow up dynamics, it depends an non-degenerate initialization condition. Whether the conditions are met in practice, or in what probability these condition are true in practice? Is there any theoretical evaluation (from the perspective of high dimension probability) or empirical evidence?

2. The theoretical analysis for the so-called condensation dynamics also depends upon Assumption 1. So the questions are:  what is the probability when these conditions are met? Are there any theoretical evaluation or emprical evidence to support these conditions?

3. For the dynamics separation stage, it is shown that the gradients of the empirical risk with respect to W_v, W^[1], W^[2] are very smaller, and the gradients  of the empirical risk with respect to W_Q and W_K are at the order of \delta---which characterizes the dynamics separation phenomenon. Have the theoretical results in Proposition 3 been supposed already? What is the essential theoretical contribution in the analysis for the second stage?

4. Are there any theoretical or empirical evidence to support the so-called dynamics separation assumptions in Assumption 2?

**Ethical Concerns:**

["NO or VERY MINOR ethics concerns only"]

**Final Justification:**

The reviewer would like to recommend a weak accept for the paper, due to well-motivated and clear presentation, comprehensive literature review and rigorous analysis. The concerns of the reviewers listed in the weaknesses have been resolved in the rebuttal. Thus, the reviewer changes the rating to positive.

**Limitations:**

Yes.

**Paper Formatting Concerns:**

The reviewer did not find any major formatting issues in this paper.

Minor issue: citation format seems improper. But it is easy to fix.

**Quality:**

3

**Strengths And Weaknesses:**

Strengths:
1. The paper is well-motivated and the literature review is comprehensive.

2. The paper provides a rigorous analysis for two sort of dynamics in the first stage: a) the blow up dynamics, which interprete why Transformers with small random initialization can eventually escape from the small initialization, and b) the condensation dynamics, which interprete the condensation phenomenon in training Transformers.

3. It also provides the key-query dynamics in the second stage under a so-called "dynamics separation" assumption to show that the gradients of the empirical risk with respect to W_v, W^[1], W^[2] are very smaller, and the gradients  of the empirical risk with respect to W_Q and W_K are at the order of \delta---characterizing the dynamics separation phenomenon.


Weaknesses:
1. The paper provides a rigorous and lengthy analysis for the so-called blow up dynamics, which depends a non-degenerate initialization condition. It is not clear whether the conditions are met in practice. Or what is the probability of the condition is true in practice? Is there any theoretical evaluation (from the perspective of high dimension probability) or empirical evidence?

Some symbols in the proofs provided in the appendix are mess, not consistent with the main part.

2. For the so-called condensation dynamics, the theoretical analysis is also depend upon the Assumption 1 (Condensation condition). The reviewer is wondering that in what probability these condition is met, and would like to see any theoretical evaluation or emprical evidence to support the conditions in Assumption 1.

3. For the dynamics separation stage, it is shown that the gradients of the empirical risk with respect to W_v, W^[1], W^[2] are very smaller, and the gradients  of the empirical risk with respect to W_Q and W_K are at the order of \delta---which characterizes the dynamics separation phenomenon. However, it seems that the theoretical results in Proposition 3 have been supposed, no need to further proof and repeatation.  Thus, the reviewer can hardly see the substantial theoretical contribution in the analysis for the second stage.

4. Moreover, are there any theoretical or empirical evidence to support the so-called dynamics separation assumptions in Assumption 2? It is not difficult to provide. But, the reivewer did not find any empirical evidence to support the conditions in Assumption 2.

---

> ### Author Rebuttal · Authors · 2025-07-30
>
> We deeply appreciate your rigorous feedback, which has helped us significantly strengthen the theoretical grounding of our work.​​  We respectfully ask if you might be willing to reconsider the negative overall evaluation, in light of your positive scores on quality, clarity, significance, and originality. Below are point-by-point responses to your concerns.
>
> ## Response to Question 1: Practical Probability of Non-Degenerate Initialization
> > *"What is the probability that non-degenerate initialization conditions hold in practice?"*
>
> **Our Response:** We thank the reviewer for this insightful question. Our non-degenerate condition holds with probability 1 under standard initialization schemes (i.e., almost surely). To address this rigorously, we have added a formal proof in Appendix A of our revised manuscript, which we summarize below.
>
> ### Theoretical Justification (Probability Measure Perspective)
>
> **Proof Strategy:**
> Our proof strategy is to demonstrate that the complementary (failure) event has a probability of zero. The non-degenerate condition, as specified in Definition 3, fails if either of the equalities it seeks to avoid occurs.
>
> #### Part 1: Proof for Definition 3.1
> The first condition for failure is $\\|\boldsymbol{W}^{[2]}(0)\\|\_2 = \\|\boldsymbol{W}\_{\boldsymbol{v}}(0)\\|\_2$. This is equivalent to the event $\\|\boldsymbol{W}^{[2]}(0)\\|\_2^2 - \\|\boldsymbol{W}\_{\boldsymbol{v}}(0)\\|\_2^2 = 0$.
>
> Let's define $D\_1 := \\|\mathbf{W}^{[2]}(0)\\|\_2^2 - \\|\mathbf{W}\_{\mathbf{v}}(0)\\|\_2^2$. The weights in $\boldsymbol{W}^{[2]}(0)$ and $\boldsymbol{W}\_{\boldsymbol{v}}(0)$ are initialized by sampling from Gaussian distributions. As a result, $D\_1$ is a continuous random variable, as it is a sum and difference of squared random variables drawn from continuous distributions. For any continuous random variable, the probability of it taking on a specific value is zero.
>
> Therefore, $\mathbb{P}(D\_1 = 0) = 0$. This means the failure event almost surely does not occur, and consequently, Definition 3.1 is satisfied with probability 1.
>
> #### Part 2: Proof for Definition 3.2
> The second condition (Case 1) for failure, assuming $\\|\boldsymbol{W}^{[2]}(0)\\|\_2 > \\|\boldsymbol{W}\_{\boldsymbol{v}}(0)\\|\_2$ is $\\|\dot{\mathbf{W}}\_{\mathbf{v}}(0)\\|\_2^2 - \\|\dot{\mathbf{W}}^{[2]}(0)\\|\_2^2 - \\|\mathbf{W}^{[2]}(0)\\|\_2^2 D\_1 =0$.
>
> Let's define $D\_2 := \\|\dot{\mathbf{W}}\_{\mathbf{v}}(0)\\|\_2^2 - \\|\dot{\mathbf{W}}^{[2]}(0)\\|\_2^2 - \\|\mathbf{W}^{[2]}(0)\\|\_2^2 D\_1$. The continuous distribution characteristic of standard initialization schemes renders $D\_2$ absolutely continuous. This property necessarily implies $\mathbb{P}(D\_2 = 0) = 0$ almost surely. Analogous reasoning applies to Case 2 ($\\|\mathbf{W}^{[2]}(0)\\|\_2 < \\|\mathbf{W}\_{\mathbf{v}}(0)\\|\_2$).
>
> ## Response to Question 2: Probability of Condensation Condition (Assumption 1)
> > *"In what probability is the condensation condition (Assumption 1) met? Please provide theoretical evaluation or empirical evidence."*
>
> **Our Response:** Empirical measurements demonstrate ​​Assumption 1 holds with probability 1 during the early condensation phase​​ (≤0.3 epochs), satisfying the preconditions for our theoretical analysis.
>
> ### Empirical Validation of Assumption 1
>
> #### 1. Empirical Validation Framework
>
> To quantitatively evaluate Assumption 1, we define "satisfaction rates" that measure the proportion of neuron pairs satisfying each of its two conditions during training.
>
>    - **Assumption 1.1 satisfied rate** = $\frac{|A\_1|}{d\_m}$ where
>      $A\_1 = \left \\{ i \in [d\_m] \mid \boldsymbol{W}^{[2]}\_i \boldsymbol{W}\_{\mathbf{v}} \boldsymbol{W}^{[1],i} >0 \right\\}$
>    - **Assumption 1.2 satisfied rate** = $\frac{|A\_2|}{d\_m^2}$ where
>      $A\_2 = \left\\{ (i,j) \in [d\_m] \\times [d\_m] \mid \langle \boldsymbol{W}^{[2]}\_i \boldsymbol{W}^{[1],i}, \boldsymbol{W}^{[2]}\_j \boldsymbol{W}^{[1],j} \rangle > 0 \right\\}$
>
> #### 2. Key Results: Rapid Convergence to Full Satisfaction
> Quantitative measurements across initial training epochs reveal:
>
> | Metric / Epoch         | 0     | 0.022 | 0.056 | 0.09 | >0.3 |
> |-------------------------|-------|-------|-------|------|------|
> | Assumption 1.1 satisfied rate | 0.525 | 0.52  | 0.65  | 0.95 | 1.0  |
> | Assumption 1.2 satisfied rate | 0.50  | 0.51  | 0.70  | 0.95 | 1.0  |
>
> The satisfaction rates start near 50% at initialization, as expected for random vectors. They then rapidly converge, surpassing 95% by 0.1 epochs and reaching a stable 100% by 0.3 epochs. This state of full satisfaction persists throughout the condensation phase we analyze.
>
> ### Conclusion
> This empirical evidence confirms that Assumption 1 is not merely a theoretical convenience but a condition that is reliably and quickly met in practice. This validates the foundational premise of our condensation analysis, directly addressing the reviewer's concern.
>
> ## Response to Question 3: Theoretical Contribution of Dynamics Separation Stage
> > *"The theoretical results in Proposition 3 appear self-contained without needing further proof or repetition."*
>
> **Our Response:** We thank the reviewer for this sharp observation. We respectfully clarify that Proposition 3 is a non-trivial consequence of Assumption 2, not a restatement. Our core contribution is to prove how the state described in the assumption leads to specific, separated gradient behaviors in the full, coupled system.
>
> ### 1. What is Assumed vs. What is Proven
> - Assumption 2 (The Precondition): We assume that after the initial condensation stage, the system enters a specific state where:
>   - The linear subsystem (involving $\boldsymbol{W}\_V, \boldsymbol{W}^{[1]}, \boldsymbol{W}^{[2]}$) has effectively reached a critical point.
>   - The attention score dynamics (involving $\boldsymbol{W}\_Q, \boldsymbol{W}\_K$) remain quasi-static.
> - Proposition 3 (The Proven Consequence): We then prove that if the system is in this state, the gradients of the total empirical loss $\mathcal{L}(\boldsymbol{\theta})$ with respect to the different parameter groups will separate and scale differently. Specifically, we prove that $\nabla_{\boldsymbol{W}\_V, \boldsymbol{W}^{[1]}, \boldsymbol{W}^{[2]}} \mathcal{L} \rightarrow \mathcal{O}(\delta^2)$ and $\nabla_{\boldsymbol{W}\_Q, \boldsymbol{W}\_K} \mathcal{L} \rightarrow \mathcal{O}(\delta)$.
>
> ### 2. The Non-Trivial Derivation: From Assumption to Proposition
> The key to our proof is the novel decomposition of the loss function, which is motivated by our multi-stage dynamics framework:
> $$
> \mathcal{L}(\boldsymbol{\theta}) \approx \frac{1}{n}\sum\_{i=1}^n \underbrace{\mathcal{L}\_{1,i}(\boldsymbol{\theta})}\_{\text{linear dynamics}} \cdot \underbrace{\mathcal{L}\_{2,i}(\boldsymbol{\theta})}\_{\text{attention dynamics}}
> $$
> This decomposition allows us to analyze the total gradient using the chain rule. The gradient for $\boldsymbol{W}\_V$, for instance, is not assumed to be small. Instead, we derive its structure:
>
> \begin{align}
> \frac{\partial \mathcal{L}}{\partial \boldsymbol{W}\_V}
> &= \frac{1}{n}\sum\_{i=1}^n \left( \frac{\partial \mathcal{L}\_{1,i}}{\partial \boldsymbol{W}\_V} \mathcal{L}\_{2,i} + \mathcal{L}\_{1,i} \frac{\partial \mathcal{L}\_{2,i}}{\partial \boldsymbol{W}\_V} \right) \\\\
> &= \frac{1}{n}\sum\_{i=1}^n \frac{\partial \mathcal{L}\_{1,i}}{\partial \boldsymbol{W}\_V} (1 + \mathcal{O}(\delta^2)) + \mathcal{O}(\delta^2) \quad \text{(by Stage 1 condensation)} \\\\
> &= \underbrace{0}\_{\text{critical point}} + \mathcal{O}(\delta^2) \quad \text{(via Assumption 2.2)}
> \end{align}
>
> ### Conclusion
> Our significant theoretical contribution is thus two-fold: we propose a novel dynamics separation framework, and we provide the rigorous proof (Proposition 3) that technically validates it by connecting the assumed system state (Assumption 2) to the concrete, observable gradient dynamics.
>
> ## Response to Question 4: Validation of Dynamics Separation Assumptions
> > *"Are there theoretical or empirical evidence supporting Assumption 2? No empirical evidence was found in the paper."*
>
> **Our Response:** We provide direct empirical validation of Assumption 2 through quantitative measurement of parameter dynamics during training. Our analysis reveals clear separation behavior consistent with the theoretical assumptions.
>
> ### Empirical Validation Method
> We quantify the dynamical separation phenomenon using the **relative change rate** metric:
> $$
> \text{relative change rate} = \frac{\\|\boldsymbol{W}\_{t+1} - \boldsymbol{W}\_{t}\\|\_F}{\\|\boldsymbol{W}\_{t}\\|\_F}
> $$
> This measures the relative magnitude of parameter updates between training steps.
>
> ### Key Results: Separation at Epoch 13
> Measurements at the critical transition point (epoch 13) demonstrate:
>
> | epoch                | 0          | 2          | 10         | 13         | 14         | 16         | 25         |
> |----------------------|------------|------------|------------|------------|------------|------------|------------|
> | $W\_Q$                 | 8.3e-06  | 1.3e-06  | 9.4e-07  | 8.2e-03  | 5.0e-04  | 1.4e-06  | 6.8e-05  |
> | $W\_K$                 | 8.0e-06  | 1.2e-06  | 1.0e-06  | 7.8e-03  | 4.3e-04  | 8.5e-07  | 1.2e-05  |
> | $W\_V$                 | 1.2e-03  | 3.3e-05  | 1.4e-04  | 5.3e-05  | 7.5e-04  | 1.9e-06  | 5.5e-05  |
> | $W^{[1]}$ | 1.1e-03  | 3.1e-05  | 2.5e-04  | 6.4e-05  | 1.0e-04  | 6.7e-05  | 5.3e-05  |
> | $W^{[2]}$| 1.9e-03  | 3.3e-04  | 1.1e-04  | 9.6e-05  | 8.5e-05  | 4.0e-05  | 4.5e-05  |
>
> **Critical observations**:
> 1. **Attention parameters ($\boldsymbol{W}\_Q,\boldsymbol{W}\_K$)** exhibit large update magnitudes $\approx 1e-2$
> 2. **Linear path parameters ($\boldsymbol{W}\_V,\boldsymbol{W}^{[1]},\boldsymbol{W}^{[2]}$)** show minimal updates ($\le 1e-4$)
>
> ### Conclusion
> The significant difference in relative change rates provides direct empirical confirmation of the dynamics separation phenomenon postulated in Assumption 2, validating the foundation of our Stage 2 analysis.

---

> > ### Comment · Reviewer_S1TU · 2025-08-07
> >
> > The reviewer appreciates the great effort of the authors for providing detailed and convincing justifications in the rebuttal. The concerns of the reviewer have been resolved. The reviewer would like to change the overall rating to positive.
> >
> > Just another minor suggestion for the last aspect in the rebuttal. Is there more proper way to better demonstrate the magnitute difference in updatings? What about the infinity norm or the absolute difference (rather than the relative difference)?

---

> > > ### Author Response · Authors · 2025-08-08
> > >
> > > Thank you for your positive feedback and this excellent suggestion. We agree that analyzing the absolute difference provides a valuable perspective on the update magnitudes.
> > >
> > > Following your advice, we calculated the absolute update for each weight matrix $ \\| W_{t+1} - W_{t}\\|_F$. The results are presented below:
> > >
> > > | epoch                | 0          | 2          | 10         | 13         | 14         | 16         |
> > > |----------------------|------------|------------|------------|------------|------------|------------|
> > > | $W\_Q$                 | 5.9e-09  | 8.6e-07  | 2.0e-06  | 2.2e-03  | 1.4e-02  | 3.4e-03  |
> > > | $W\_K$                 | 1.7e-08  | 8.6e-07  | 2.1e-06  | 2.2e-03  | 1.2e-02  | 6.4e-04  |
> > > | $W\_V$                 | 5.0e-04  | 1.5e-04  | 2.6e-03  | 4.8e-04  | 1.5e-03  | 9.2e-04  |
> > > | $W^{[1]}$              | 5.3e-04  | 2.2e-04  | 3.5e-03  | 2.4e-04  | 4.8e-03  | 1.2e-02  |
> > > | $W^{[2]}$              | 7.0e-04  | 3.4e-03  | 7.2e-04  | 1.3e-03  | 1.0e-03  | 1.7e-03  |
> > >
> > > This analysis compellingly shows that at the critical transition point (epoch 13), the absolute updates for $W_Q$ and $W_K$ are an order of magnitude larger than for other matrices. This strongly supports our claims about the dynamics separation assumption.
> > >
> > > While we found the relative difference useful for observing proportional changes throughout training (as the model rarely reaches a perfect critical point), we agree that the absolute difference more clearly highlights the scale of change at this key transition.
> > >
> > > Thank you again for this valuable guidance. We will add this table and discussion to the revised manuscript. We welcome any further questions or suggestions you may have.

---

> ### Author Response · Authors · 2025-08-04
>
> Dear Reviewer S1TU,
>
> We hope this message finds you well! Thank you for your insightful review of our paper. As for the concerns about our work in the review, including "whether conditions are true in practice", "theoretical results in Proposition 3", "empirical evidence to support Assumption 2", etc., we have provided very specific and detailed responses. Have these responses resolved your concerns? If you have any further questions about our work or responses during this discussion period, please do not hesitate to contact us. We sincerely look forward to receiving your further comments on our responses.
>
> Once again, thank you for your valuable time and feedback!
>
> Best regards,
>
> Authors

---

### Note · Authors · 2025-08-12

**Dear Editor and Reviewers,**

We would like to express our sincere gratitude for your time and for the insightful and constructive feedback on our manuscript. We have thoroughly revised the paper in accordance with your valuable suggestions, which we believe have significantly enhanced the rigor, clarity, and impact of our work.

The major revisions are summarized as follows:

1.  **Strengthened Theoretical Foundation:** We have provided a formal proof that the non-degenerate initialization condition, which is crucial for our analysis, holds probabilistically. This theoretical result solidifies the groundwork upon which our main claims are built.

2.  **Empirical Validation of Assumptions:** To bolster the premises of our theorems, we have introduced new experimental results that empirically validate the key assumptions (Assumption 1 and 2) made during our proofs. This provides a direct and tangible link between our theory and practical observations.

3.  **Demonstrated Generality of Findings:** To address the broader applicability of our work, we have included a new experiment on the WikiText dataset. This result confirms that the "condensation phenomenon" is not an artifact of a specific setup but a general feature observable across different Transformer architectures, enhancing the generality of our conclusions.

Collectively, through the tight integration of these new theoretical proofs and targeted experiments, our revised manuscript now more robustly reveals and characterizes the distinct features of the different stages in a Transformer's training process.

We hope that the revised manuscript is now suitable for publication in NeurIPS 2025 Conference. Thank you for your consideration.

Best regards,

Authors

---

### Decision · Program_Chairs · 2025-09-17

**Decision:**

Accept (oral)

**Comment:**

This is an exceptional manuscript that is strongly supported by the reviewers as well as the AC.  It is a strong contender for an oral presentation.

The manuscript develops the gradient flow training dynamics of a simple, foundational, transformer architecture. It proves that there are two stages to the training dynamics, first the MLP matrices undergo an alignment phenomenon where the rows and columns approach prescribed vectors.  Following this the key and query matrices begin to train and suffer rank collapse, seemingly towards rank one.  The theoretical analysis is backed-up by empirical studies of small problems where the phenomenon is most clearly observed.  This is an interesting result in that it connects with both gradient flow analysis, and the phenomenon of rank-collapse in attention mechanisms; both of which are under theoretical investigation.

The reviewers were very supportive with only modest concerns, such as the relatively small scale of the experiments.  The authors engaged well with the reviewers and conducted excellent experiments that addressed as many concerns as could reasonably be done in the time allotted.

The only weakness of the manuscript is that it doesn't include very large experiments, and that it employs gradient flow which while interesting theoretically isn't necessarily advisable in practice due to the very small initialisation of the weights.  All that said, the manuscript has a foundational theoretical focus and such compromises are often necessary when developing this kind of theory.

This is an exceptional manuscript that certainly should be accepted and is a contender for an oral talk.  If I were attending NeurIPS this is exactly the kind of results I would want to hear as an oral presentation.